# Mutational phospho-mimicry reveals a regulatory role for the XRCC4 and XLF C-terminal tails in modulating DNA bridging during classical non-homologous end joining

**Davide Normanno[1], Aurélie Négrel[1†], Abinadabe J de Melo[1], Stéphane Betzi[1], Katheryn Meek[2,3]\*, Mauro Modesti[1]\***

[1]Cancer Research Center of Marseille, CNRS UMR7258, Inserm U1068, Institut Paoli-Calmettes, Aix-Marseille Université UM105, Marseille, France; [2]Department of Microbiology and Molecular Genetics, Michigan State University, East Lansing, United States; [3]Department of Pathobiology and Diagnostic Investigation, College of Veterinary Medicine, Michigan State University, East Lansing, United States

**Abstract** XRCC4 and DNA Ligase 4 (LIG4) form a tight complex that provides DNA ligase activity for classical non-homologous end joining (the predominant DNA double-strand break repair pathway in higher eukaryotes) and is stimulated by XLF. Independently of LIG4, XLF also associates with XRCC4 to form filaments that bridge DNA. These XRCC4/XLF complexes rapidly load and connect broken DNA, thereby stimulating intermolecular ligation. XRCC4 and XLF both include disordered C-terminal tails that are functionally dispensable in isolation but are phosphorylated in response to DNA damage by DNA-PK and/or ATM. Here we concomitantly modify the tails of XRCC4 and XLF by substituting fourteen previously identified phosphorylation sites with either alanine or aspartate residues. These phospho-blocking and -mimicking mutations impact both the stability and DNA bridging capacity of XRCC4/XLF complexes, but without affecting their ability to stimulate LIG4 activity. Implicit in this finding is that phosphorylation may regulate DNA bridging by XRCC4/XLF filaments.

**\*For correspondence:** kmeek@ msu.edu (KM); mauro.modesti@ inserm.fr (MM)

**Present address:** [†]Sigma Aldrich, Saint Quentin Fallavier, France

**Competing interests:** The authors declare that no competing interests exist.

## Introduction

In mammalian cells, classical non-homologous end joining (c-NHEJ) is the primary pathway that repairs DNA double-strand breaks (DSBs). This pathway is specialized in the repair of two-ended DSBs like those that are induced by ionizing radiation or by many radiomimetic drugs. The c-NHEJ pathway is also exploited by developing lymphocytes to repair DSBs that are introduced by the RAG1/2 endonuclease during antibody and T-cell receptor V(D)J gene rearrangement. The c-NHEJ machinery is versatile and adapts to efficiently repair DSBs with diverse DNA end structures (*Lieber, 2010*; *Reid et al., 2017*). Core c-NHEJ effectors include the heterodimeric Ku DNA end sensor, the DNA-dependent protein kinase catalytic subunit (DNA-PKcs), DNA Ligase 4 (LIG4), and the structurally related XRCC4 and XLF proteins. Current dogma advocates a sequential model for c-NHEJ as follows: (1) Ku performs the recognition step, rapidly and avidly binding and protecting broken DNA ends; (2) Ku recruits DNA-PKcs and promotes DNA end synapsis, regulating DNA end access; (3) if required, DNA end processing is carried on by non-core factors (Artemis, X-family

**eLife digest** DNA in human and other animal cells is organised into structures called chromosomes. One of the most dangerous types of DNA damage is a double-strand break, where both strands of the DNA helix are broken in the same place. If this damage is not repaired it can be serious enough to kill the cell. If the DNA is repaired badly, part of one chromosome can become attached to another – a defect known as a chromosome translocation.

Fortunately, cells are equipped with machineries that can recognise and fix these breaks. One of these processes is known as "non-homologous end joining" and it involves a set of proteins including two known as XRCC4 and XLF. These proteins work like a bandage, holding together the broken DNA until it is repaired. Both proteins have long tails, but the role of these structures was not clear.

During DNA repair, the cell chemically modifies the tails of these proteins by a process called phosphorylation. However, previous studies have found that it is possible to prevent the modification of the tail of one of the proteins, or even remove the tail entirely, without affecting the repair process. Here, Normanno et al. investigated the effect of blocking the modification of the tails of both proteins at the same time.

For the experiments, the tails were both altered in various places to either block or mimic the phosphorylation that normally occurs during DNA repair. Mimicking the phosphorylation of both tails affected the ability of XRCC4 and XLF to stay attached to the DNA, suggesting that the phosphorylation helps these proteins to detach from the DNA once the repair is complete. Furthermore, in human embryonic kidney cells the altered proteins were less able to repair DNA damage in response to a drug that causes double-strand breaks.

These findings improve our understanding of how cells repair their DNA to maintain a complete set of genetic information. Defects in DNA repair are linked to conditions where the brain does not develop properly, whilst some cancer therapies deliberately inflict double-strand breaks to kill cancer cells. In the future, these findings may lead to improvements in radiotherapy and other treatments for human diseases.

polymerases, or other DNA modifying enzymes); and 4] the LIG4/XRCC4 complex, assisted by XLF, catalyzes the ligation step directly.

Emerging data are challenging this sequential model, and suggest a more dynamic and adaptive model for c-NHEJ. These paradigm shifting data include structural, biochemical, and functional studies revealing that LIG4, XRCC4 and XLF form filamentous structures that bridge DNA and promote ligation in vitro (*Andres et al., 2012*; *Hammel et al., 2011*; *Ropars et al., 2011*; *Roy et al., 2015*, *2012*; *Wu et al., 2011*). This DNA tethering, which has been observed in vitro, implies an early function for XRCC4/XLF filaments during DNA repair; we have proposed that XRCC4/XLF complexes rapidly and robustly 'splint' broken DNA as 'sleeve-like' protein bandages (*Brouwer et al., 2016*). In fact, a recent study showed that in living cells, XRCC4 and XLF, independently of LIG4, form filamentous super-structures at bleomycin-induced DSBs that adopt various configurations (*Reid et al., 2015*). Currently it is unclear exactly how these filaments function in the context of c-NHEJ; however, specifically disrupting XRCC4-XLF filament formation (by mutagenesis) affects c-NHEJ function in living cells (*Roy et al., 2015*). Thus, addressing the dynamics and regulation of XRCC4-XLF super-structures is an important area of investigation with regard to a better understanding of the c-NHEJ mechanism.

Here we focus on understanding the functional contribution of the XRCC4 and XLF C-terminal regions, which have so far been considered to be dispensable for c-NHEJ. XRCC4 and XLF both have disordered C-terminal tails that are phosphorylated by DNA-PK and/or ATM in response to DSBs [of note, numerous studies have reported substantial hyper-phosphorylation of both XRCC4 and XLF in response to genotoxic stress in living cells]. Early functional studies (using phospho-blocking and phospho-mimetic mutants) suggested that these phosphorylations do not alter the efficacy or fidelity of c-NHEJ in living cells (*Yu et al., 2003*, *2008*). However, potential functional consequences of these phosphorylations might be masked in studies focusing on either XRCC4 or XLF

individually, if phosphorylation of these sites is functionally redundant, as it has been shown for many DNA-PK dependent phosphorylations (*Cui et al., 2005*; *Ding et al., 2003*; *Neal et al., 2014*). Thus, here we utilize biochemical and cell-based approaches to address the impact of XRCC4 and XLF phospho-blocking or phospho-mimicking mutations in concert. Whereas, blocking or mimicking eight phosphorylation sites in XRCC4 and 6 sites in XLF does not abate stimulation of LIG4 in vitro, nor diminish end-joining of episomal substrates in living cells, phospho-mimicking dramatically affects XRCC4/XLF dissociation from DNA and stability of DNA tethering in vitro, and also impedes cellular survival after exposure to radio-mimetic drugs, especially drugs that induce complex DNA lesions. Moreover, phospho-mimicking mutations alter repair of chromosomal DNA DSBs in living cells, resulting in increased nucleotide loss and greater dependence on end-terminal short sequence homologies.

## Results

### The C-terminal tails of XRCC4 and XLF are important to form super-complexes in vitro that bridge DNA favoring molecular collision rates that enhance DNA ligation

The phosphorylation of XRCC4 and XLF has been addressed by several investigators. Briefly, we identified nine phosphorylation sites in XRCC4 (*Yu et al., 2003*); Lees-Miller and colleagues identified six sites in XLF (*Yu et al., 2008*). Studies using phospho-specific antibodies and numerous phospho-proteome reports have confirmed that eight of the nine sites (residues S193, S260, S304, S315, S320, T323, S327, and S328 – using UniProt Q13426 isoform 1 as reference) in XRCC4 and minimally four of the six sites (residues S132, S203, S245 and S251 – using UniProt Q9H9Q4 isoform 1 as reference) in XLF are phosphorylated in vivo (http://www.phosphosite.org). Remarkably, all the above-mentioned phosphorylation sites are located in the C-terminal tails of XRCC4 and XLF (*Figure 1*).

To test whether the disordered C-terminal tails of XRCC4 and XLF are implicated in regulation of the DNA interaction dynamics of XRCC4-XLF complexes, phospho-mimetic (Asp) and phospho-ablating (Ala) variants of XRCC4 and XLF targeting the multiple DNA-PK or ATM dependent phosphorylation sites were designed and generated (*Figure 1—figure supplement 1*). To study the impact of these modifications, several different activity assays were optimized and validated using wild-type (WT) XRCC4 and XLF proteins (*Figure 2*). These assays include electrophoretic-mobility shift assays (EMSAs, *Figure 2A and B*), phage T4 DNA ligase assays (as a way to indirectly measure DNA bridging activity, *Figure 2C,D and E*), and protein-mediated DNA pull-down assays to directly measure DNA bridging activity (*Figure 2F and G*). Overall, analyses of the activity of the WT proteins confirmed that XRCC4 and XLF cooperatively form super-complexes detectable by EMSA (*Figure 2A and B*) that strongly promote intermolecular DNA cohesive and blunt end ligation (*Figure 2C,D and E*), by favoring molecular collision rates during ligation through a robust DNA bridging activity as can be seen in *Figure 2F and G*. In the latter experiment, it can be also observed that XLF has an intrinsic DNA bridging activity that depends on the presence of its C-terminal tail (*Figure 2G*, lanes 3 and 5, XLF (1-224) lacks the 75 C-terminal residues). This intrinsic XLF DNA bridging activity is strongly stimulated by XRCC4 (*Figure 2G*, lanes 3 and 7) in a manner that depends on physical interaction between XLF and XRCC4, as tested using an XLF mutant (L115D) described previously that does not interact with XRCC4 but has an intact C-terminal tail (*Figure 2G*, lanes 4 and 8, *Malivert et al., 2010*; *Roy et al., 2015*). Furthermore, as seen in *Figure 2—figure supplement 1*, the importance of the XRCC4 and XLF C-terminal tails, and especially XLF's tails in promoting DNA bridging and increasing the rate of molecular collision during ligation in vitro, is demonstrated using the truncated tail-less XRCC4(1–157) and XLF(1-224) variants.

### Phospho-mimicking XRCC4 disrupts XRCC4 DNA binding, but formation of DNA/XRCC4/XLF super-complexes is not disrupted by phospho-mimicking modifications in the C-terminal disordered tails of either XRCC4 or XLF

We first analyzed XRCC4 and XLF phosphorylation site mutants for DNA binding by EMSA (*Figure 3*). Neither the intrinsic DNA binding of XRCC4 nor its ability to form super-complexes with XLF-WT is affected by phospho-blocking eight sites in XRCC4 (*Figure 3A*, XRCC4-Ala). In contrast,

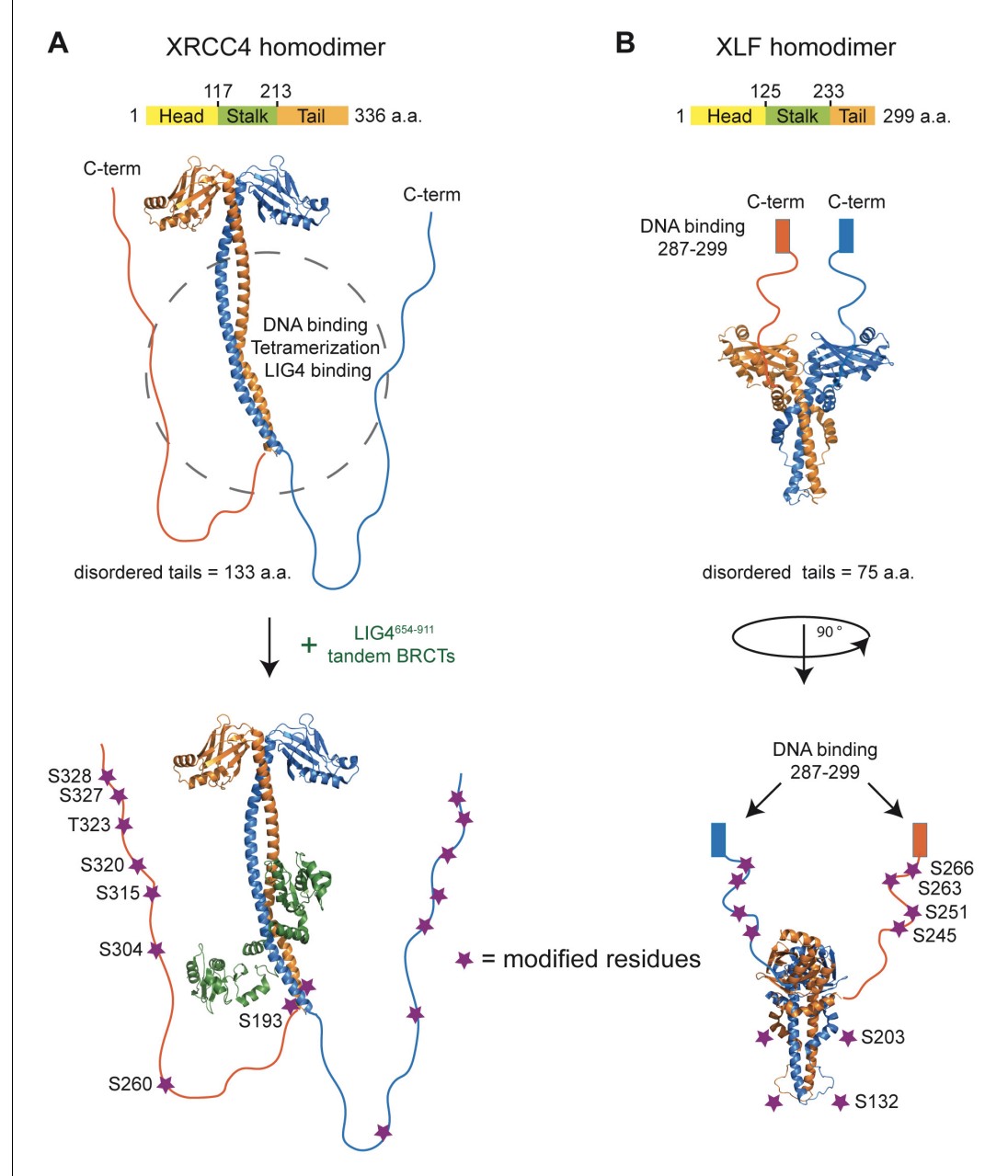

**Figure 1.** Structural organization of XRCC4 and XLF homodimers. (**A**) Structure of XRCC4 homodimer based on PDB 3II6, including residues 1–203 of XRCC4 and a schematic representation of the C-terminal disordered tails. In the bottom panel, also residues 654–911 of LIG4 are shown (green). (**B**) Structure of XLF homodimer based on PDB 2R9A, including residues 1–224 and a schematic representation of the C-terminal tails. Purple stars indicate the residues modified for phospho-mimicking or -blocking, and all were mutated, respectively, to Asp or Ala in this study. Residues are numbered using as reference Uniprot Q13426 isoform 1 and Q9H9Q4 isoform 1 for XRCC4 and XFF, respectively.

The following figure supplement is available for figure 1:

**Figure supplement 1.** Coomassie stained reducing SDS-PAGE analysis of recombinant XRCC4 and XLF variants produced in bacteria (1 µg/lane).

phospho-mimicking these eight sites in XRCC4 strongly reduces XRCC4's intrinsic DNA binding and modestly reduces cooperative DNA binding with XLF-WT (***Figure 3A***, XRCC4-Asp). Neither

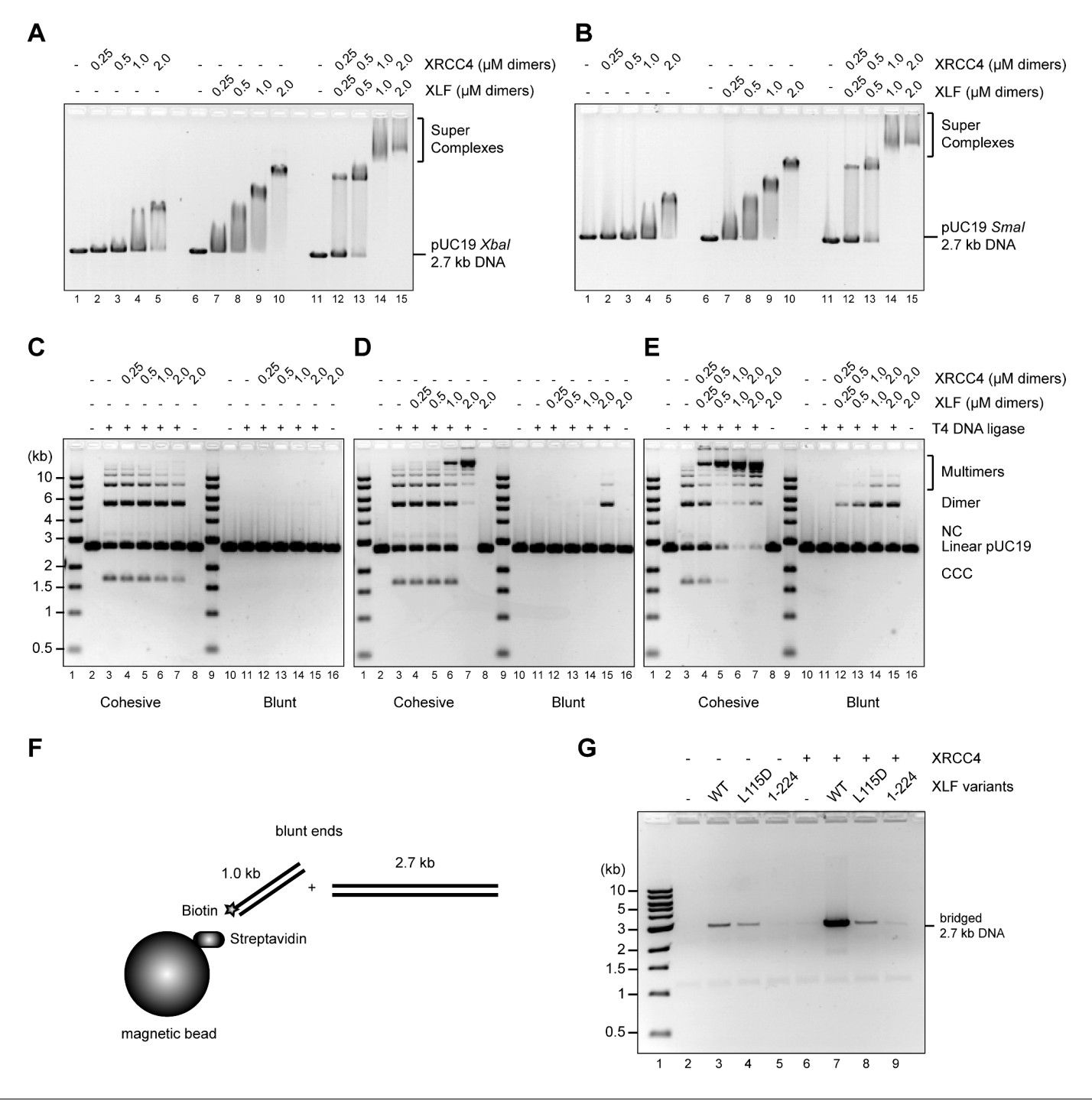

**Figure 2.** Wild-type XRCC4 and XLF form super-complexes in vitro that bridge DNA favoring molecular collision rates that enhance DNA ligation. (**A**) EMSA using a 2.7 kb DNA fragment with protruding ends and the indicated proteins resolved by agarose gel electrophoresis and DNA detected by ethidium bromide staining. (**B**) EMSA using a blunt-ended 2.7 kb DNA fragment and the indicated proteins resolved by agarose gel electrophoresis and DNA detected by ethidium bromide staining. (**C**) (**D**) (**E**) T4 DNA ligase assay using a 2.7 kb DNA fragment with cohesive (left) or blunt ends (right) and the indicated proteins. Ligation products were deproteinized and resolved by agarose gel electrophoresis followed by detection by ethidium bromide staining. NC = nicked circle, CCC = covalently closed circle. (**F**) Schematic of the DNA pull-down/bridging assay. A one-end biotinylated 1 kb DNA fragment is first attached to streptavidin-coated magnetic beads and then incubated with a blunt-ended 2.7 kb DNA fragment in the presence of the proteins of interest. (**G**) Bridging assays for the indicated proteins resolved by agarose gel electrophoresis after deproteinization followed by DNA detection by ethidium bromide staining.

*Figure 2 continued on next page*

*Figure 2 continued*

The following figure supplement is available for figure 2:

**Figure supplement 1.** T4 DNA ligase assay using a 2.7 kb DNA fragment with cohesive (left) or blunt ends (right) at three different protein equimolar concentrations.

phospho-blocking nor phospho-mimicking six sites in XLF alters either the intrinsic DNA binding activity of XLF or its ability to cooperatively form super-complexes with XRCC4-WT (*Figure 3B*).

Protein-protein cross-linking experiments in the absence of DNA suggest that the XRCC4 and XLF variants interact at the protein-protein level similarly to WT XRCC4 and XLF (*Figure 4*). To further investigate these interactions, we used Isothermal Titration micro-Calorimetry (ITC) to quantify the thermodynamic parameters of the XRCC4-XLF protein-protein interaction in the absence of DNA. Experiments were performed at 37°C (after optimization tests at 10, 25, and 37°C) because this condition provided the highest heat signature and optimal signal to noise ratio (*Figure 5* and *Figure 5—figure supplement 1*). Two experimental replicas were performed in duplicate using two protein concentrations (*Table 1*). The dissociation constant values ($K_d$) obtained for the three different combinations (WT-WT, Ala-Ala, and Asp-Asp) are equal within the experimental errors, and are approximately 2 µM (*Figure 5D*, top panel, and *Table 1*). Nevertheless, the XRCC4 and XLF variants have different enthalpic ($\Delta H^0$, dark grey bars) and entropic ($-T\Delta S^0$, light grey bars) contributions (*Figure 5D*, bottom panel, and *Table 1*) that compensate one another resulting in similar free energies ($\Delta G^0$, black bars). We conclude that although the phospho-blocking and -mimicking mutations in the disordered tails do not affect the overall equilibrium of the XRCC4-XLF interaction, these mutations affect the thermodynamics of the interaction. Moreover, the disordered tails may contribute to stabilizing the XRCC4-XLF interaction. In fact, the $K_d$ measured with full-length (WT) proteins is about half of that measured for XRCC4 and XLF truncations missing the disordered tails (*Malivert et al., 2010*). Also, ITC experiments showed that for all variants (and conditions) the stoichiometry of the XRCC4 and XLF interaction is 1:1 (*Table 1*). In sum, ITC experiments confirm that, in the absence of DNA, the Ala and Asp variants interact in solution with similar affinity than WT XRCC4 and XLF, despite the fact that the underlying molecular interactions have different thermodynamic contributions. We suggest that this may indicate phosphorylation induced conformational changes (see Discussion).

## Aspartate substitutions ablate DNA bridging

To assess the role of the XRCC4 and XLF C-terminal tails, we first used T4 ligase assays to indirectly infer DNA bridging by XRCC4-XLF complexes. The XRCC4-Ala variants combined with either XLF-WT or XLF-Ala variants modestly enhanced ligation as compared to the WT counterpart (*Figure 6*, lanes 5, 6, 8, and 9) suggesting more stable DNA-protein complexes. In contrast, stimulation of T4 ligase activity is minimal with the Asp mutants; this is most dramatic when aspartate mutants of both XRCC4 and XLF are utilized concomitantly (*Figure 6*, lanes 13). The DNA bridging activity of XRCC4 and XLF variants was also investigated via a protein-mediated DNA pull-down assay (*Figure 7*). When both XRCC4-Asp and XLF-Asp variants are combined, DNA bridging activity is severely impaired (*Figure 7*, lanes 18). Of note, the ability of the XLF-Asp variant to bridge long DNA on its own is also strongly reduced (*Figure 7B*, lane 8) although its ability to bind to DNA is not affected when tested by EMSA (*Figure 3B*). From these data, we conclude that XRCC4-Asp and XLF-Asp variants are defective in cooperative DNA bridging, even though the proteins interact with one another analogously (i.e. with similar $K_d$) to WT XRCC4 and XLF. We therefore hypothesized that XRCC4 and XLF with aspartate substitutions in the C terminal tails might generate complexes that dissociate faster from DNA as compared to WT XRCC4-XLF complexes.

## Aspartate substitutions in the disordered C-terminal tails of XRCC4 and XLF accelerate dissociation of XRCC4-XLF complexes from DNA

To assess dissociation rates of XRCC4-XLF complexes from DNA, a Surface Plasmon Resonance (SPR) assay was established. One-ended biotinylated 400 bp-long dsDNA molecules were captured

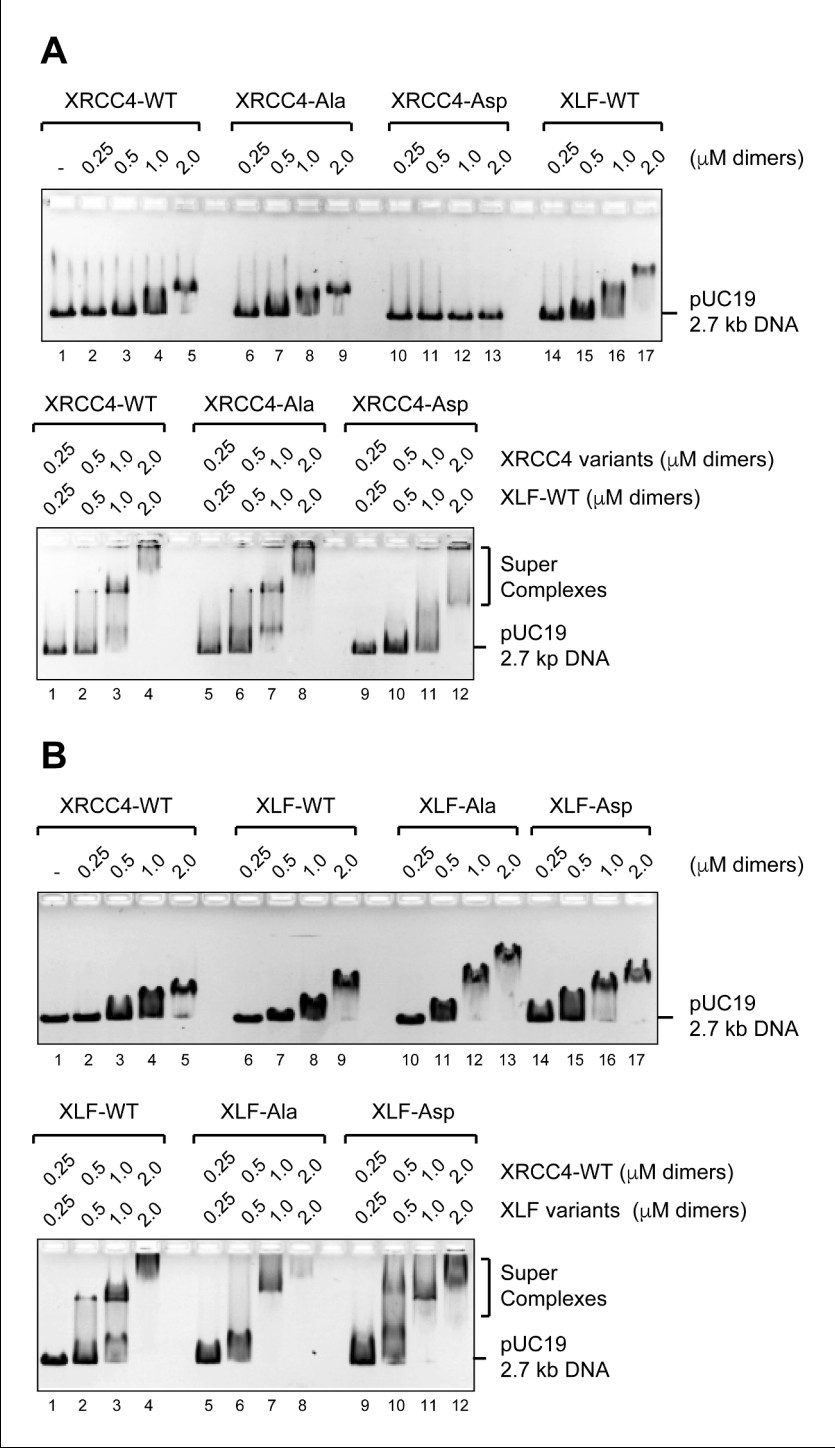

**Figure 3.** Formation of DNA/XRCC4/XLF super-complexes is not disrupted by phosphorylation site mutation. (**A**) EMSAs showing DNA binding of XRCC4 variants in isolation (top panel) and cooperatively with XLF-WT (bottom panel) resolved by agarose gel electrophoresis and detected by ethidium bromide staining. (**B**) EMSAs showing DNA binding of XLF variants in isolation (top panel) and cooperatively with XRCC4-WT (bottom panel) resolved by agarose gel electrophoresis and detected by ethidium bromide staining.

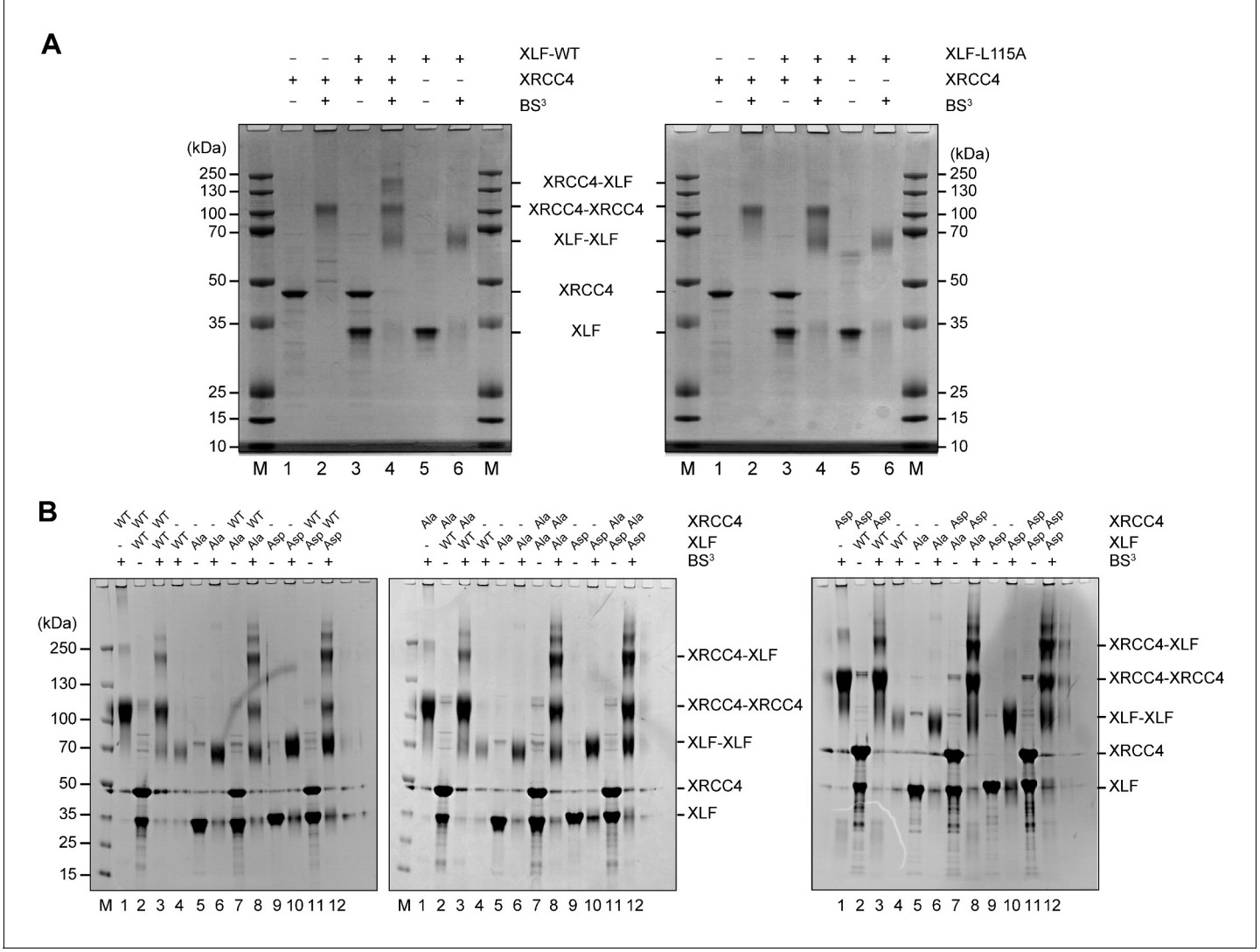

**Figure 4.** Phosphorylation site mutations do not alter XRCC4-XLF direct interaction. XRCC4 and XLF were cross-linked in isolation or in equimolar combination with BS$^3$. Cross-linked species were resolved by reducing and denaturing SDS-PAGE and detected by Coomassie staining. (**A**) XRCC4 and XLF WT-WT combination (left panel) and XRCC4-WT combined with XLF-L115A, that has a point mutation that diminished affinity for XRCC4 (right panel). (**B**) XRCC4-WT (left panel), XRCC4-Ala (middle panel) and XRCC4-Asp (right panel) respectively cross-linked to all the XLF variants. M = protein molecular weight size ladder.

on streptavidin-coated chips; protein/DNA association and dissociation was monitored using a Biacore instrument (*Figure 8A*). WT proteins were first characterized in isolation and at equi-molar concentrations (*Figure 8B*). We observed that XRCC4-WT efficiently binds DNA only when flushed at high concentrations, but once bound, the protein remains stably attached to DNA (*Figure 8B*, left panel). As compared to XRCC4, XLF's association with DNA is much faster, but XLF also dissociates faster from DNA than XRCC4 (*Figure 8B*, middle panel). When WT XRCC4 and XLF are mixed together in solution, at an equimolar ratio, and flushed, a strong enhancement of DNA association was observed followed by dissociation with comparable kinetics to XLF alone (*Figure 8B*, right panel). Probably as a consequence of polymerization of the filamentous structures formed by XRCC4 and XLF on the DNA, the SPR association and dissociation curves could not be perfectly fitted with a simple 1:1 Langmuir model to compute first order association and dissociation rate constants ($k_{on}$ and $k_{off}$, respectively). Nevertheless, we estimate that XRCC4-WT's association rate is at least 10-times slower than that of XLF-WT alone or that of the XRCC4-WT + XLF-WT combination. Moreover,

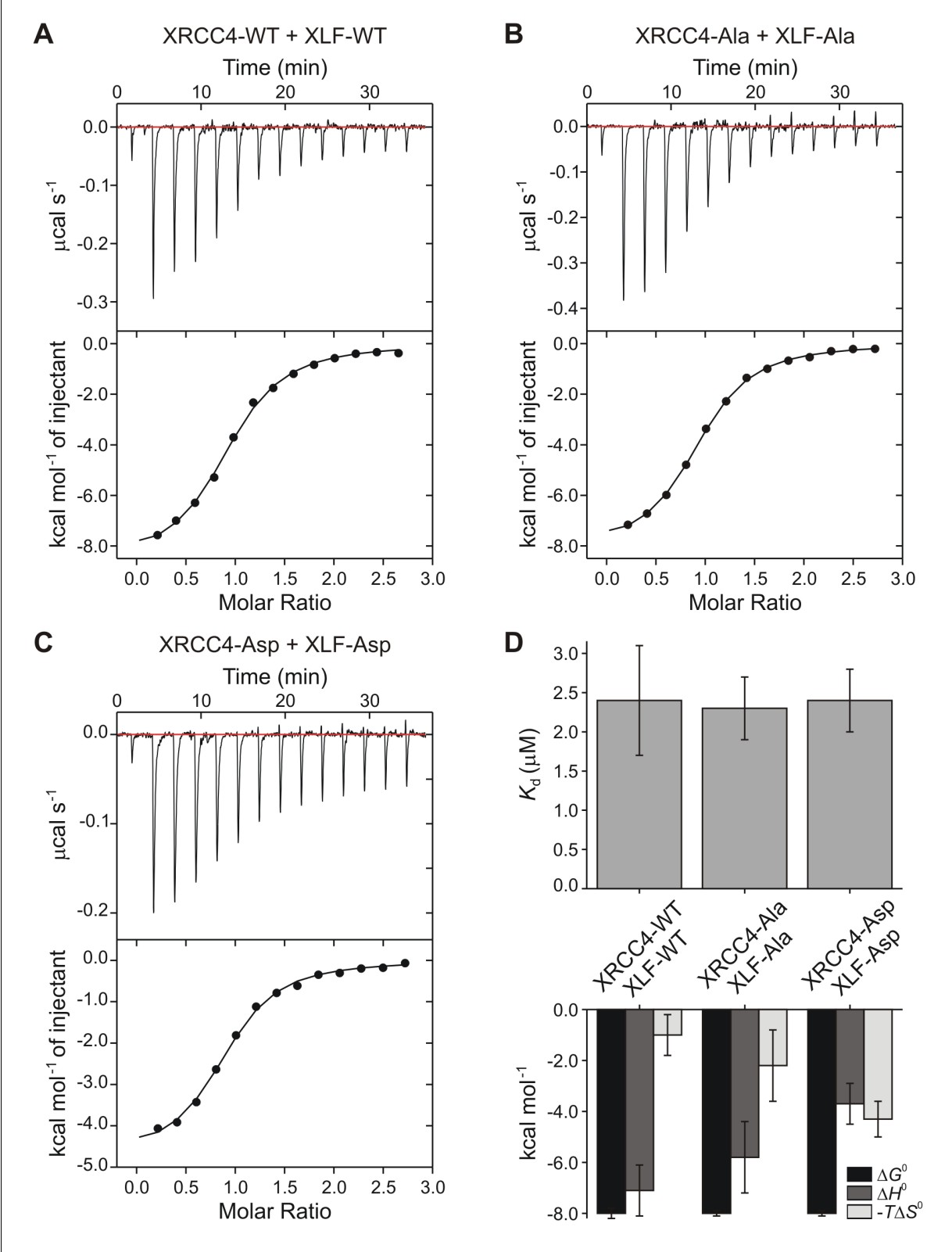

**Figure 5.** Thermodynamic analysis of wild-type and phospho-mimetic XRCC4-XLF interaction. Isothermal titration micro-calorimetry data obtained at 37°C injecting (**A**) XLF-WT (154 μM) into XRCC4-WT (12 μM), (**B**) XLF-Ala (250 μM) into XRCC4-Ala (19 μM), (**C**) XLF-Asp (250 μM) into XRCC4-Asp (19 μM). Top panels display the thermogram of the heat exchanged as a function of time after baseline (red lines) subtraction. Bottom panels show the result of the (numerical) integration of the peaks as a function of the molar ratio between the titrant (XLF) and the analyte (XRCC4); black lines are non-

*Figure 5 continued on next page*

*Figure 5 continued*

linear least squares fits of the data. Data are representative results out of two experimental replicas each composed of a technical duplicate (see *Table 1* for details). Source data are included in *Table 1—source data 1*. (D) Equilibrium constant ($K_d$) (top) and thermodynamic parameters (free energy $\Delta G^0$, black bars; enthalpy $\Delta H^0$, dark grey bars; and entropy $-T\Delta S^0$, light grey bars) (bottom) obtained from sigmoidal fit (black lines in bottom panels of (A, B, C) of ITC curves. Bars show mean values and error bars represent the standard error of the mean (SEM) of the four experimental runs performed per each condition. Thermodynamic parameter values are reported in *Table 1*.

The following source data and figure supplements are available for figure 5:

**Figure supplement 1.** Isothermal titration micro-calorimetry thermograms.

**Figure supplement 1—source data 1.** Thermograms of control experiments.

also XRCC4's dissociation rate is at least 10-times slower than that of XLF-WT alone or in combination with XRCC4-WT. When the XRCC4 and XLF Asp and Ala variants were analyzed in isolation at a fixed concentration (1 μM), association and dissociation kinetics were strongly affected for XRCC4-Ala, XRCC4-Asp and XLF-Asp but just minority for XLF-Ala (*Figure 8—figure supplement 1A*). Next, given the above-mentioned limits in correctly quantifying the association and dissociation rate constants, we decided to systematically analyze all the possible combinations of the different variants at a fixed concentration ([$C$] = 1 μM each, *Figure 8C*). To inspect and compare the behavior of the different combinations during the association phase, we considered the equilibrium affinity (defined as the response unit values averaged over 5 s taken 3 s before the start of the dissociation phase, gray band in *Figure 8C*), which reports on the ratio: [$C$]·$k_{on}$ / $k_{off}$. The XRCC4-Ala and XLF-Ala variants together, or in combination with the WT variant, exhibited equilibrium affinities similar to the WT-WT combination (*Figure 8D*). The XRCC4-Asp and XLF-Asp variants combined with the WT and Ala variants also showed modest deviations from the WT-WT combination. In contrast, the combination of the two Asp variants resulted in a 4-fold drop in the equilibrium affinity as compared to the WT-WT combination. To understand the observed drop for the XRCC4-Asp + XLF-Asp complexes, we analyzed the dissociation phase. The dissociation curves of all combinations have a biphasic character, with an initial rapid dissociation followed by a slower dissociation phase (*Figure 8—figure supplement 1B*). The slower dissociation kinetic component ($k_{off\_}slow$) was similar for all combinations of XRCC4 and XLF variants, but, interestingly, the initial rapid dissociation phase ($k_{off\_}fast$) was accelerated 4-fold in the case of the XRCC4-Asp + XLF-Asp complexes as compared to the WT-WT complexes (*Figure 8E*). Thus, the decrease in equilibrium affinity detected for the XRCC4-Asp + XLF-Asp complexes can be explained by the increase in the rate of dissociation without a major change in the rate of association of XRCC4-Asp + XLF-Asp complexes. Taken together, these analyses indicate that simultaneous aspartate substitutions within the C-terminal tails of both XRCC4 and XLF results in a shifted DNA binding equilibrium, such that complexes bind to DNA similarly to WT XRCC4/XLF complexes, but detach much faster. This likely explains the impairment in DNA bridging, as assessed by pull-down and T4 DNA end ligation assays. Lastly, SPR experiments using C-terminal tail-less variants XRCC4(1–157) and XLF(1-224) confirmed that C-terminal tails are required (and especially those of XLF) for stable DNA binding (*Figure 8—figure supplement 1C and D*), consistently with DNA bridging results obtained with the T4 DNA ligation assays (*Figure 2—figure supplement 1*).

## XRCC4 and XLF phospho-mimetics do not bridge DNA but are sufficient in stimulating LIG4-XRCC4 activity

XRCC4 is in large excess over LIG4 (*Mani et al., 2010*). On the one hand XRCC4 associates with XLF to form complexes that bridge broken DNA during c-NHEJ (*Brouwer et al., 2016*), and on the other hand, as a separate function, it associates with LIG4 to form a 2:1 XRCC4/LIG4 complex that performs the ligation step of c-NHEJ, which is further stimulated by XLF (*Lu et al., 2007*; *Tsai et al., 2007*). Therefore, to assess the role of the XRCC4 and XLF C-terminal tails in this distinct function of XRCC4, we next considered whether the XRCC4 and XLF Ala and Asp phosphorylation site mutants could support LIG4 function. Recombinant LIG4-XRCC4 complexes were produced incorporating XRCC4-WT, XRCC4-Ala or XRCC4-Asp (*Figure 9—figure supplement 1*); and these complexes

**Table 1.** Impact of phospho-mimicking or phospho-blocking mutations on the thermodynamics parameters of XRCC4-XLF protein-protein interaction by micro-calorimetry.

| | T (°C) | 2-Me (mM) | [XRCC4] (µM) ITC cell | [XLF] (µM) ITC syringe | Ratio | N | $K_a$ (M$^{-1}$) | $\Delta H^0$ (kcal/mol) | $K_d$ (µM) | $-\Delta G^0$ (kcal/mol) | $-T\Delta S^0$ (kcal/mol) | Thermogram shown in | Source data | $<K_d>$ (µM) | $<H^0>$ (kcal/mol) | $<-\Delta G^0>$ (kcal/mol) | $<-T\Delta S^0>$ (kcal/mol) |
|---|---|---|---|---|---|---|---|---|---|---|---|---|---|---|---|---|---|
| XRCC4-WT XLF-WT | 10 | 10 | 17 | 168 | 10 | n.d. | n.d. | n.d. | n.d. | n.d. | n.d. | Figure 5—figure supplement 1A | A | | | | |
| | 25 | 10 | 17 | 168 | 10 | 0.97 | 536000 | −1.7 | 1.8 | −7.8 | −6.1 | – | B | 3.2 ± 0.8 | −2.4 ± 0.4 | −7.5 ± 0.2 | −5.2 ± 0.5 |
| | 25 | 10 | 33 | 330 | 10 | 0.91 | 306000 | −2.9 | 3.2 | −7.5 | −4.6 | Figure 5—figure supplement 1B | C | | | | |
| | 25 | 10 | 25 | 330 | 13 | 0.84 | 222000 | −2.6 | 4.5 | −7.3 | −4.7 | – | D | | | | |
| | 37 | 0 | 23 | 300 | 13 | 0.85 | 273000 | −6.0 | 3.7 | −7.7 | −1.7 | – | E | 2.4 ± 0.7 | −7.1 ± 1 | −8.0 ± 0.2 | −1.0 ± 0.8 |
| | 37 | 0 | 23 | 300 | 13 | 0.89 | 278000 | −4.9 | 3.6 | −7.7 | −2.9 | – | F | | | | |
| | 37 | 0 | 12 | 154 | 13 | 0.92 | 848000 | −8.6 | 1.2 | −8.4 | 0.2 | Figure 5A | G | | | | |
| | 37 | 0 | 12 | 154 | 13 | 0.84 | 760000 | −8.8 | 1.3 | −8.3 | 0.4 | – | H | | | | |
| XRCC4-Ala XLF-Ala | 25 | 10 | 25 | 330 | 13 | 1.21 | 300000 | −1.7 | 3.3 | −7.5 | −5.8 | Figure 5—figure supplement 1C | I | | | | |
| | 37 | 0 | 25 | 325 | 13 | 1.07 | 637000 | −3.1 | 1.6 | −8.2 | −5.1 | – | J | 2.3 ± 0.4 | −5.8 ± 1.4 | −8.0 ± 0.1 | −2.2 ± 1.4 |
| | 37 | 0 | 25 | 325 | 13 | 1 | 343000 | −3.7 | 2.9 | −7.8 | −4.1 | – | K | | | | |
| | 37 | 0 | 19 | 250 | 13 | 0.92 | 345000 | −8.4 | 2.9 | −7.8 | 0.6 | – | L | | | | |
| | 37 | 0 | 19 | 250 | 13 | 0.92 | 582000 | −8.1 | 1.7 | −8.2 | −0.1 | Figure 5B | M | | | | |
| XRCC4-Asp XLF-Asp | 25 | 10 | 35 | 350 | 10 | 1.27 | 672000 | −0.9 | 1.5 | −7.9 | −7.0 | – | N | | | | |
| | 25 | 10 | 25 | 330 | 13 | 0.82 | 263000 | −1.2 | 3.8 | −7.4 | −6.2 | Figure 5—figure supplement 1D | O | | | | |
| | 37 | 0 | 32 | 416 | 13 | 1.06 | 289000 | −2.4 | 3.4 | −7.8 | −5.4 | – | P | 2.4 ± 0.4 | −3.7 ± 0.8 | −8.0 ± 0.1 | −4.3 ± 0.7 |
| | 37 | 0 | 32 | 416 | 13 | 0.95 | 425000 | −2.3 | 2.4 | −8.0 | −5.7 | – | Q | | | | |
| | 37 | 0 | 19 | 250 | 13 | 0.88 | 566000 | −4.7 | 1.8 | −8.2 | −3.5 | Figure 5C | R | | | | |
| | 37 | 0 | 17 | 220 | 13 | 0.85 | 538000 | −5.5 | 1.8 | −8.1 | −2.6 | – | S | | | | |

2-Me stands for 2-Mercaptoethanol

Ratio indicates the titrant over analyte concentration ([XLF]/[XRCC4])

Source data specifies the column corresponding to each thermogram in **Table 1—source data 1**.

Last four columns report mean values ± Standard Error of the Mean (SEM) from four experimental runs at 37°C (grey cells) and 3 runs at 25°C (white cells)

n.d. denotes no detectable heat transfer (at 10°C)

**Source data 1.** Thermograms of XRCC4-XLF WT-WT, Ala-Ala, and Asp-Asp experiments.

were systematically analyzed for cohesive end ligation activity in combination with XLF-WT, XLF-Ala, or XLF-Asp (*Figure 9A,B and C*). As can be seen, alanine or aspartate modifications have no major impact on LIG4 activity, even when LIG4-XRCC4-Asp was combined with XLF-Asp. These data suggest that the disordered C-terminal tails of XRCC4 and XLF regulate the DNA bridging activity of XRCC4-XLF complexes, but do not directly affect the ability of XLF to stimulate the XRCC4/LIG4 complex.

## XRCC4 and XLF phosphorylation site mutations do not affect end joining of episomal substrates

Our biochemical analyses suggest that XRCC4 and XLF C-terminal tails may be redundant with one another. For example, the Asp variants of either XRCC4 or XLF paired with WT (and Ala) partners bind and dissociate from DNA analogously to the WT-WT combinations (*Figure 8*), but when the

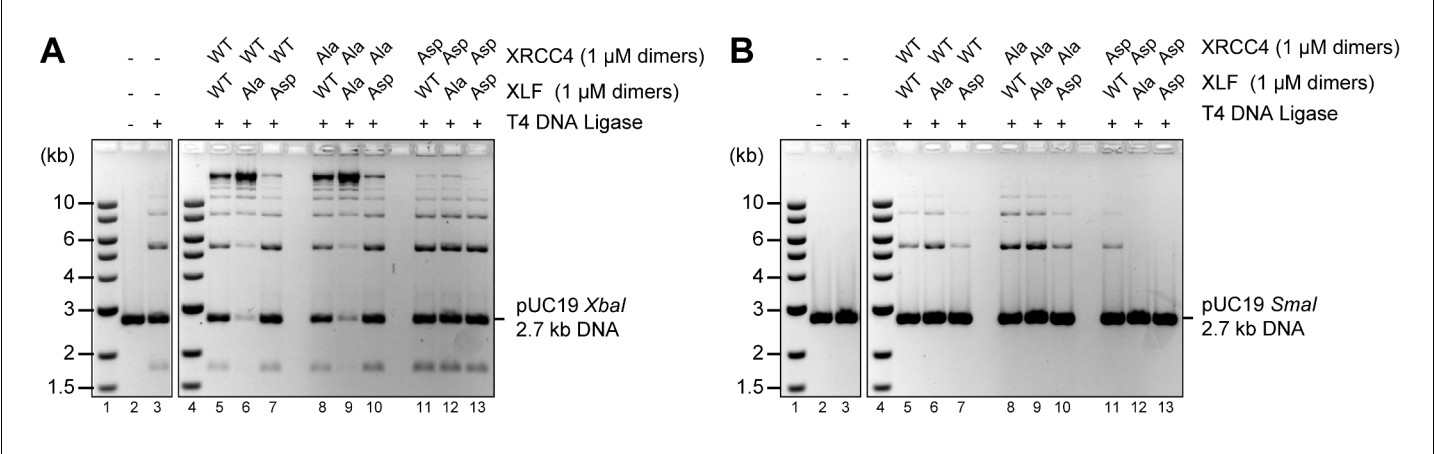

**Figure 6.** Blocking XRCC4 and XLF phosphorylation sites enhances DNA bridging; phospho-mimicking mutations abate DNA bridging. All combinations of XRCC4 variants with XLF variants tested in ability to stimulate T4 DNA ligase cohesive end ligation (**A**) or blunt end ligation (**B**). Ligation products were deproteinized and resolved by agarose gel electrophoresis followed by detection by ethidium bromide staining.

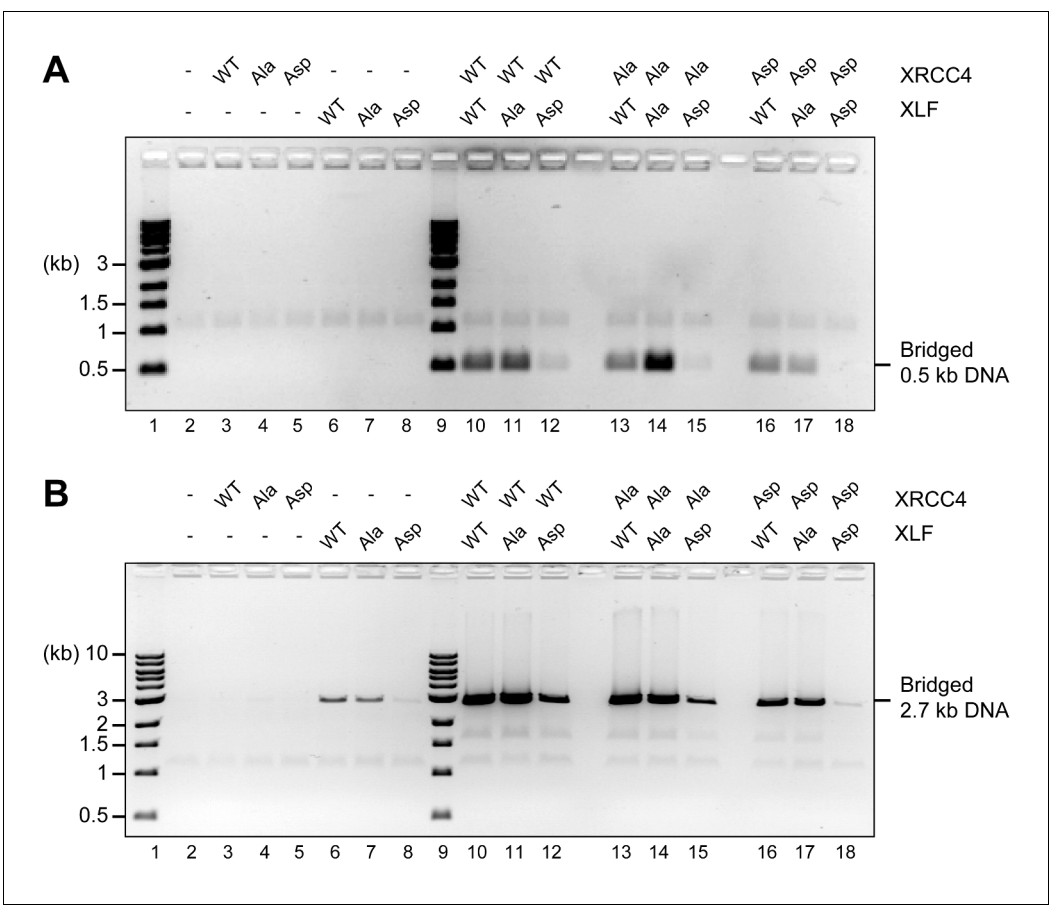

**Figure 7.** Blocking XRCC4 and XLF phosphorylation sites enhances DNA bridging; phospho-mimicking mutations abate DNA bridging. DNA bridging assays using a short blunt ended 0.5 kb DNA (**A**) or a long blunt ended 2.7 kb DNA (**B**) and the indicated proteins. Samples were deproteinized and resolved by agarose gel electrophoresis followed by detection by ethidium bromide staining.

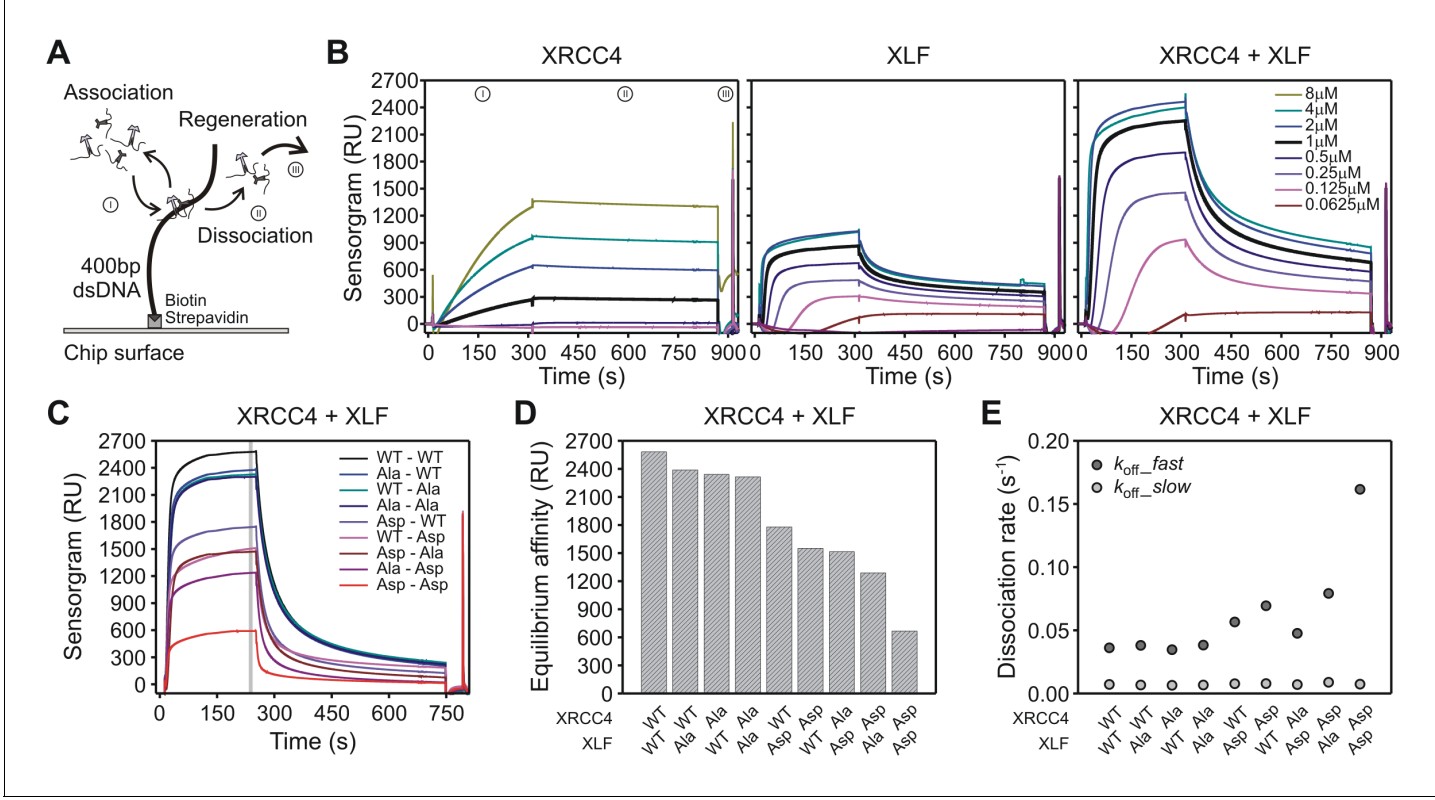

**Figure 8.** Phospho-mimicking XRCC4 and XLF mutants associate with DNA, but dissociate rapidly. (**A**) Scheme of the SPR assay with 400 bp blunt end DNA anchored to the streptavidin coated chip via biotin-streptavidin linkage. (**B**) From left to right, sensorgrams of XRCC4-WT, XLF-WT, and equi-molar mix of XRCC4-WT and XLF-WT at the indicated concentrations. Source data are provided in *Figure 8—source data 1*, *2* and *3*, respectively. (**C**) Sensorgrams of all XRCC4 and XLF variant combinations each at 1 μM concentration. Source data are provided in *Figure 8—source data 4*. (**D**) Equilibrium affinity of the different combinations of XRCC4 and XLF variants. Source data are provided in *Figure 8—source data 5*. (**E**) Dissociation rates for the different combinations of XRCC4 and XLF variants. $k_{off\_fast}$ (dark grey symbols) and $k_{off\_slow}$ (light grey symbols) correspond, respectively, to the initial (fast) and late (slow) phase in the biphasic dissociation curves. Error bars fall within symbol dimension. Source data are provided in *Figure 8—source data 5*.

The following source data and figure supplements are available for figure 8:

**Source data 1.** Sensorgrams of XRCC4 WT tritation.
**Source data 2.** Sensorgrams of XLF WT tritation.
**Source data 3.** Sensorgrams of XRCC4-XLF WT-WT tritation.
**Source data 4.** Sensorgrams of all XRCC4-XLF variant combinations.
**Source data 5.** Equilibrium affinity and dissociation rates of all XRCC4-XLF variant combinations.
**Figure supplement 1.** Sensorgrams of XRCC4 and XLF variants and truncations.
**Figure supplement 1—source data 1.** Sensorgrams of XRCC4 and XLF variants in isolation.

**Figure supplement 1—source data 2.** Sensorgrams and mono- and double-exponential fits of XRCC4-XLF WT-WT and Asp-Asp combinations.

**Figure supplement 1—source data 3.** Sensorgrams of XRCC4-XLF full-length and truncations in isolation and in combination.

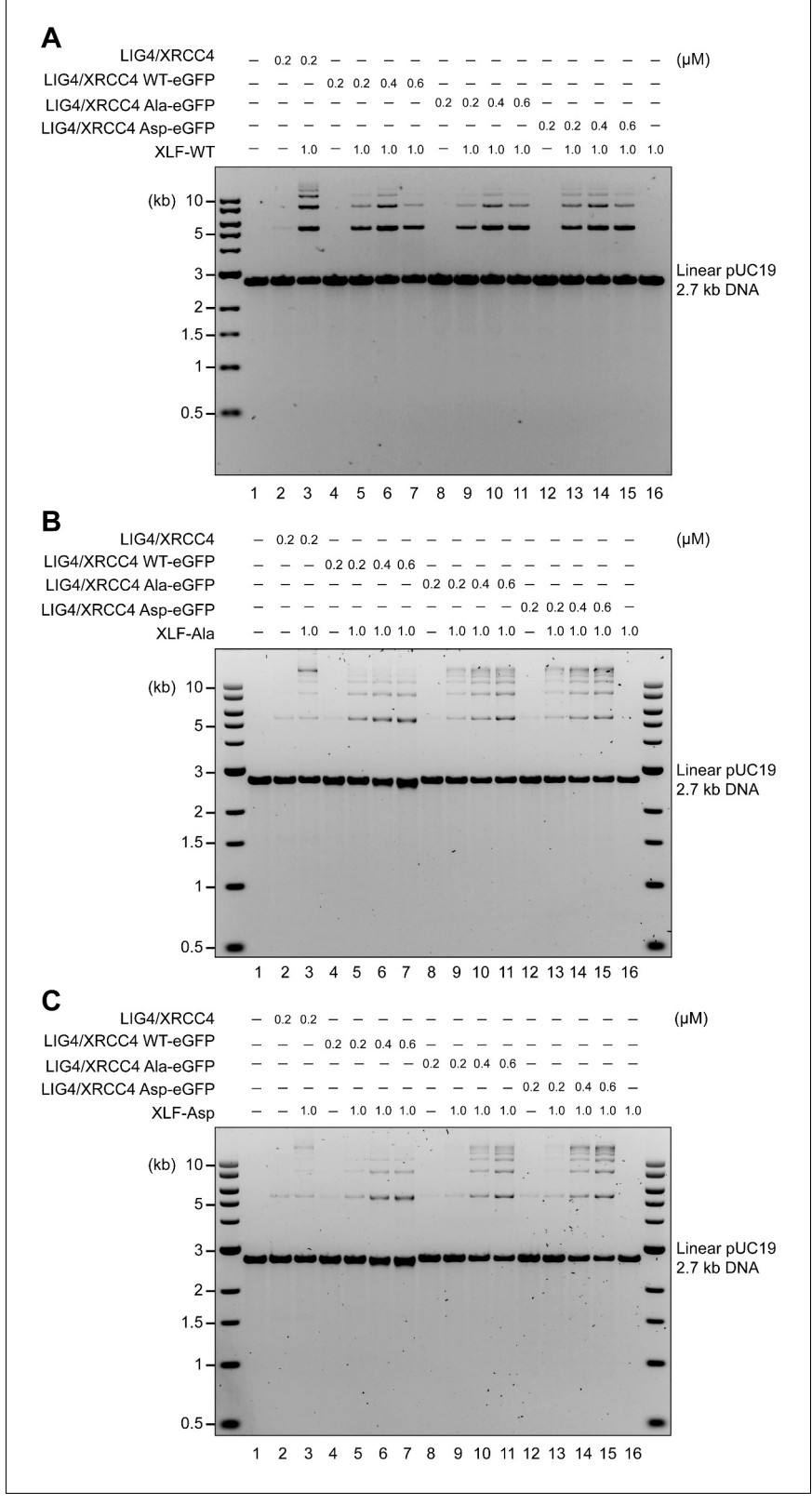

**Figure 9.** XRCC4 and XLF phospho-mimetics do not bridge DNA ends but are completely sufficient in stimulating LIG4-XRCC4 activity. Stimulation of LIG4/XRCC4-WT, LIG4/XRCC4-WT-eGFP, LIG4/XRCC4-Ala-eGFP or LIG4/XRCC4-Asp-eGFP cohesive end ligation by XLF-WT (**A**), XLF-Ala (**B**) or XLF-Asp (**C**). Ligation products were deproteinized and resolved by agarose gel electrophoresis followed by detection by ethidium bromide staining. *Figure 9 continued on next page*

*Figure 9 continued*

The following figure supplement is available for figure 9:

**Figure supplement 1.** Coomassie stained reducing SDS-PAGE analysis of recombinant LIG4/XRCC4 variants purified after overexpression in bacteria (1 μg/lane).

Asp variants are combined together, DNA dissociation is markedly more rapid. Thus, any functional impact of the C-terminal tails in living cells might be masked by this redundancy when XRCC4 or XLF are studied individually. To facilitate functional analyses of XRCC4 and XLF C-terminal disordered tails in concert in living cells, we used 293 T cells where genes encoding both proteins were disrupted using CRISPR/Cas9 (*Neal et al., 2016*). To co-express WT or mutant forms of XRCC4 and XLF, expression constructs were prepared so that C-terminal myc-tagged XRCC4 and XLF (interrupted by an IRES sequence) are expressed from the same promoter. Constructs were prepared encoding XRCC4 alone, or XRCC4 and XLF together, with all 14 DNA-PK/ATM phosphorylation sites intact, or with all sites substituted to Ala or Asp.

We have recently described a panel of episomal substrates (*Neal et al., 2016*) in which two pairs of V(D)J recombination signal sequences (RSS) or I-Sce1 sites are separated by the red fluorescent protein (RFP) coding sequence (*Figure 10*, top panels). These substrates provide an internal control (RFP expression) for plasmid uptake and nuclear localization in the transfected cells. Additionally, they include the SV40 origin of replication, and are thus efficiently replicated episomally in 293 T cells. If recombination occurs, the RFP gene is deleted, juxtaposing the CFP gene with the promoter allowing CFP expression as a reporter of end joining activity that can be quantified by flow cytometry.

Although XLF is dispensable for V(D)J recombination in murine lymphocytes, XLF is required for V(D)J recombination in human lymphocytes, and patients with mutations in XLF exhibit a SCID phenotype (*Buck et al., 2006*; *Woodbine et al., 2014*). Thus, in 293 T cells lacking both XRCC4 and XLF, transfection of the XRCC4 only encoding construct does not reconstitute either coding or signal

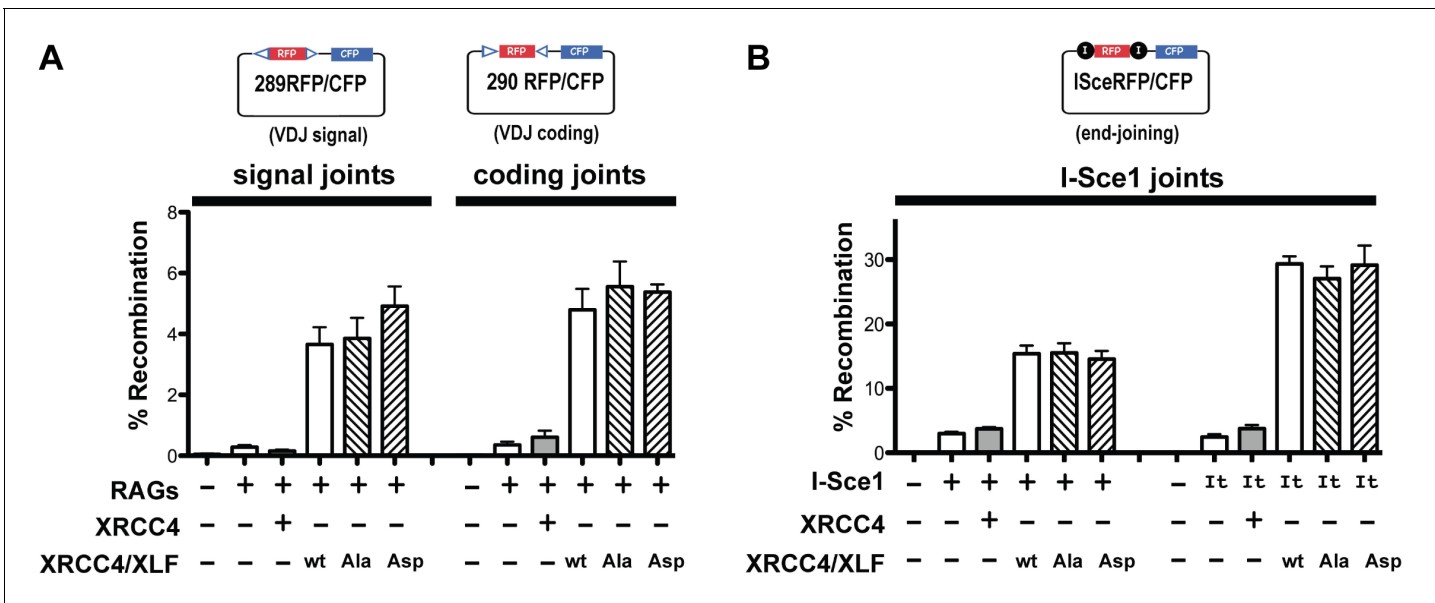

**Figure 10.** XRCC4 and XLF phosphorylation site mutation does not affect end joining of episomal substrates. Fluorescent substrates (depicted in top panels) were utilized to detect V(D)J coding and signal joints in XRCC4/XLF-deficient 293 T cells transiently expressing full-length RAG1, RAG2, and WT, Ala, or Asp mutants of XRCC4 and XLF (**A**), or joining of I-Sce1 induced DSBs (**B**). Bottom panels show percent recombination of episomal fluorescent substrates. Error bars indicate SEM from five independent experiments.

joining; in contrast, substantial complementation is observed for both coding and signal end joining when both XRCC4 and XLF are expressed (*Figure 10A*). As can be seen, the expression constructs encoding XRCC4-XLF with 14 phosphorylation sites substituted to either Ala or Asp fully reconstitute both V(D)J coding and signal end joining. We conclude that phospho-mimetic or -blocking mutations in the disordered C-terminal tails of XRCC4 and XLF do not affect joining of RAG-induced DSBs of episomal substrates.

In contrast to RAG induced DSBs that are exclusively dependent on c-NHEJ for repair, DSBs introduced by restriction enzymes can be repaired by a-NHEJ, although this repair is slower, less efficient, and requires terminal short sequence homology (*Deriano and Roth, 2013*). As can be seen in *Figure 10B*, some end joining is observed in cells lacking both XRCC4 and XLF, or in cells complemented with only XRCC4; this reflects a-NHEJ activity. However, end joining is much more efficient when both XRCC4 and XLF are expressed. This is true whether I-Sce1 alone initiates the DSBs, or whether a fusion protein of I-Sce1 and the trex exonuclease is used to initiate DSBs. As with RAG-induced DSBs, blocking or phospho-mimicking the 14 sites does not affect the efficiency of end joining. All together from these data, we conclude that stable XRCC4/XLF DNA bridging is not required for efficient end joining of proximal I-Sce1 or RAG induced DSBs on episomal substrates.

## XRCC4 and XLF complexes do not promote a-NHEJ

XRCC4 is required not only to stimulate LIG4, but is also required for LIG4 stability. An additional construct was prepared in which three point mutations were introduced into XRCC4 that disrupt XRCC4's interaction with LIG4 [F180D +I181D+A183V] (*Modesti et al., 2003*). XRCC4-XLF expressed from this construct should be capable of forming complexes, but would not stabilize or stimulate LIG4. This construct was tested for its ability to promote end joining of the I-Sce1 substrate. As can be seen (*Figure 11*), the LIG4 mutant XRCC4-XLF construct does not enhance end joining. These data suggest that DNA bridging by XRCC4-XLF complexes does not promote a-NHEJ.

## XRCC4 and XLF phosphorylation site mutants do not fully reverse sensitivity to radiomimetic drugs

To further assess the concomitant role of XRCC4 and XLF C-terminal disordered tails in the response to radiomimetic drugs, we generated stable 293 T cell clones expressing WT, Ala, Asp, or vector control by integrating the IRES co-expression constructs described above. Two clones of each construct were analyzed, but only one clone presented. As can be seen, stably reconstituted XRCC4/XLF -/- cells express levels of XRCC4 and XLF (WT and variants) similar to endogenous levels (*Figure 12A*). The complemented XRCC4 and XLF proteins have slightly slower electrophoretic mobility because of their C-terminal myc tags. The Asp mutant has an even more substantial change in electrophoretic mobility because of the charge difference that results from S/T to Asp substitution

The substantial phosphorylation of XRCC4 induced by DNA damage results in a dramatic change in electrophoretic mobility. To determine whether the phospho-blocking mutant substantially blocks damage-induced phosphorylation, XRCC4 phosphorylation was assessed by immunoblotting. As can be seen (*Figure 12B*), a marked shift in XRCC4's electrophoretic mobility is induced by zeocin (at 1 or 24 hr post treatment) in cells expressing WT but not Ala XRCC4 and XLF. The shift is equivalent to the change in mobility observed in the XRCC4 Asp mutant.

The radiomimetic antibiotics zeocin (a bleomycin analog) and neocarzinostatin induce DSBs by free-radical attack of sugar residues in both DNA strands. The DSBs that result from either drug have complex DNA ends, with the complexity of damage being more dramatic with neocarzinostatin as compared to zeocin (reviewed in *Povirk, 2012*). As can be seen (*Figure 12C and D*), WT XRCC4 and XLF substantially reverse both zeocin and neocarzinostatin sensitivity of 293T XRCC4/XLF deficient cells. In 293 T cells, blocking all 14 phosphorylation sites with alanine substitutions reduces cell survival after exposure to either zeocin or neocarzinostatin compared to cells expressing WT controls. In contrast, the Asp variants combination does not reverse either zeocin or neocarzinostatin sensitivity in XRCC4/XLF deficient 293 T cells. These data suggest that in 293 T cells, XRCC4-XLF stable DNA bridging (abrogated in the Asp mutants combination) is required for repair of zeocin and neocarzinostatin induced DSBs. Moreover, the fact that the Ala variants combination is also impaired suggests that disassembly of XRCC4-XLF complexes is also required to facilitate repair.

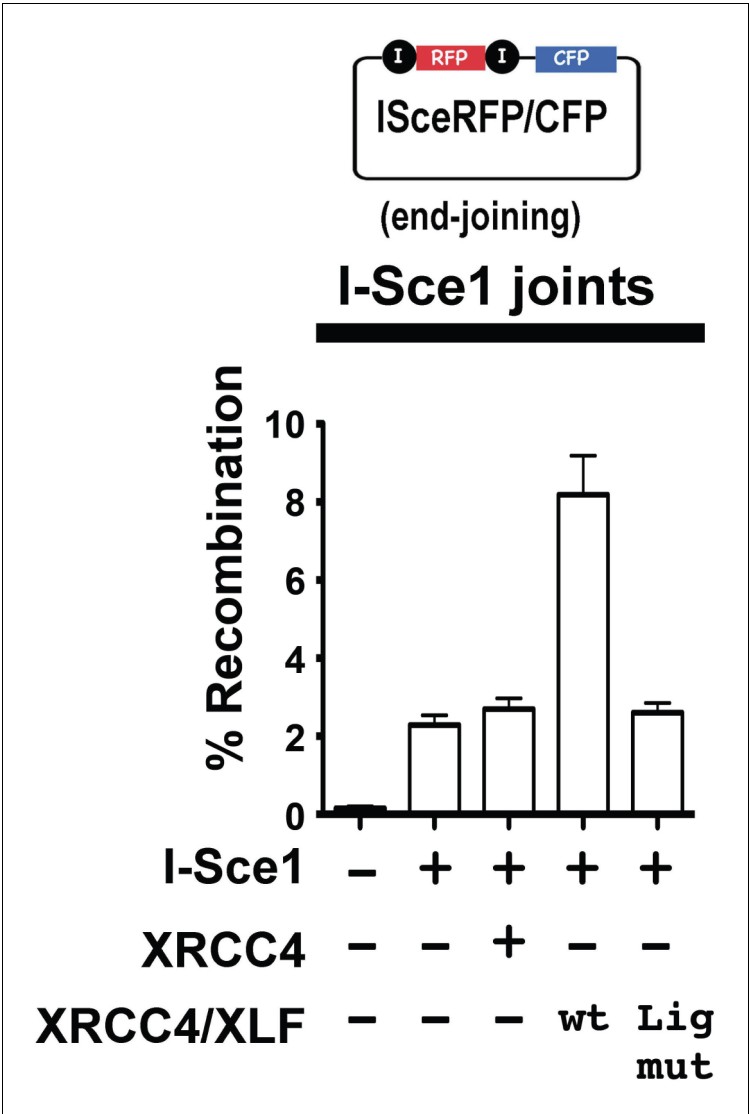

**Figure 11.** XRCC4 and XLF complexes do not promote a-NHEJ. The fluorescent substrate (depicted in the top panel) was utilized to detect joining of I-Sce1 induced DSBs. Bottom panel shows percent recombination of episomal fluorescent I-Sce1 end joining substrate in XRCC4/XLF-deficient 293 T cells transiently expressing I-Sce1, XRCC4 alone, WT XRCC4 and XLF, or an XRCC4 mutant that is defective in LIG4 interaction and WT XLF as indicated. Error bars indicate SEM from three independent experiments.

To further extend our analysis in other cell lines, we used CRISPR/Cas9 to disrupt XLF in XRCC4 deficient HCT116 cells (*Figure 12—figure supplement 1*, left panel) that were generated by AAV gene targeting (*Fattah et al., 2014*). In HCT116 cells, both the Ala and Asp variants fully reconstitute zeocin resistance (not shown). In contrast, cells expressing either Asp or Ala mutants are more sensitive to neocarzinostatin (the drug inducing more dramatic end aberrations) than WT controls (*Figure 12—figure supplement 1*, right panel). Thus, as we and others have shown previously (*Li et al., 2008*; *Roy et al., 2015*), different organisms, and different cell types have variable dependence on XLF. To corroborate these findings, we used again CRISPR/Cas9 to disrupt XLF in the XRCC4 deficient CHO cell strain XR-1 (*Figure 12—figure supplement 2*, left panel). XR-1 XRCC4/ XLF double deficient cells were transfected with the XRCC4-XLF expression constructs. Similar, to 293 T cells, whereas WT XRCC4 and XLF substantially reverse neocarzinostatin hypersensitivity, complementation is progressively less substantial with the Ala and Asp variant constructs (*Figure 12— figure supplement 2*, right panel).

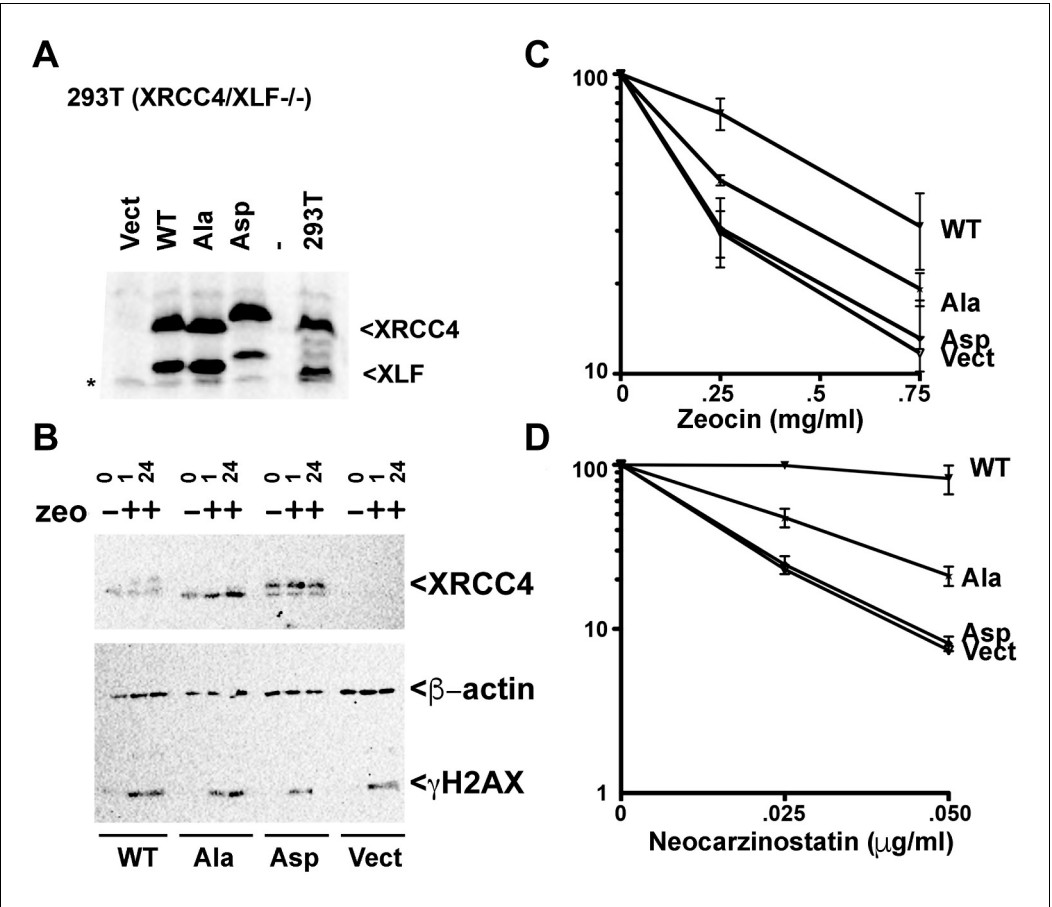

**Figure 12.** XRCC4 and XLF phosphorylation site mutants do not fully reverse sensitivity to radio-mimetic drugs. (A) Immunoblots of stable expression of WT, Ala, or Asp mutants of XRCC4 and XLF in 293 T cells that lack both XRCC4 and XLF. * non-specific species. (B) 293 T cells expressing WT, Ala, Asp, or no XRCC4 or XLF were exposed to Zeocin (500 μg/ml for 1 hr) or not and analyzed after 1 hr or 24 hr by immunoblotting for XRCC4, $\beta$-actin, or γ-H2AX as indicated. Zeocin (C) and Neocarzinostatin (D) sensitivity of 293T XRCC4/XLF double deficient cell strains complemented by stable expression of equivalent levels of WT, Ala, or Asp mutants of XRCC4 and XLF. Error bars indicate SEM from at least three independent experiments.

The following figure supplements are available for figure 12:

**Figure supplement 1.** XRCC4 and XLF phosphorylation site mutants do not fully reverse sensitivity to radio-mimetic drugs.

**Figure supplement 2.** XRCC4 and XLF phosphorylation site mutants do not fully reverse sensitivity to radio-mimetic drugs.

## Phospho-mimicking XRCC4/XLF alters repair of chromosomal DNA DSBs

The in vitro analyses of XRCC4/XLF phospho-mimetic complexes reported above demonstrated that phospho-mimicking ablates DNA bridging but does not alter XRCC4/XLF's ability to promote LIG4 activity. We have also shown previously (*Roy et al., 2015*) that a separation of function XLF mutant, proficient in stimulating LIG4 but deficient in DNA bridging, is functional in supporting episomal V(D)J joining, consistently with the phospho-mimetic mutants presented here, which are also proficient in stimulating LIG4 but deficient in DNA bridging. We next addressed whether joining of chromosomal DSBs is affected by disrupting XRCC4/XLF mediated DNA bridging.

To assess chromosomal end joining in isogenic cell strains expressing WT, Ala, or Asp versions of XRCC4 and XLF, we exploited CRISPR/Cas9 targeting of the FANCG gene on chromosome 9, which we found to be remarkably efficient in 293 T cells. We reasoned that DNA bridging might be required specially to join distant DSBs, and targeting reagents for three sites within ~4 kb on chromosome nine were prepared (*Figure 13A*). Joining of two separate deletions was assessed, one that results in ~300 bp deletion, and one that results in ~4 kb deletion. Although this strategy does not allow quantification of joining, the quality of joining can be assessed by sequencing. 293 T cells expressing WT XRCC4 and XLF, 14 X Ala, 14 X Asp, or no XRCC4/XLF were transfected with various

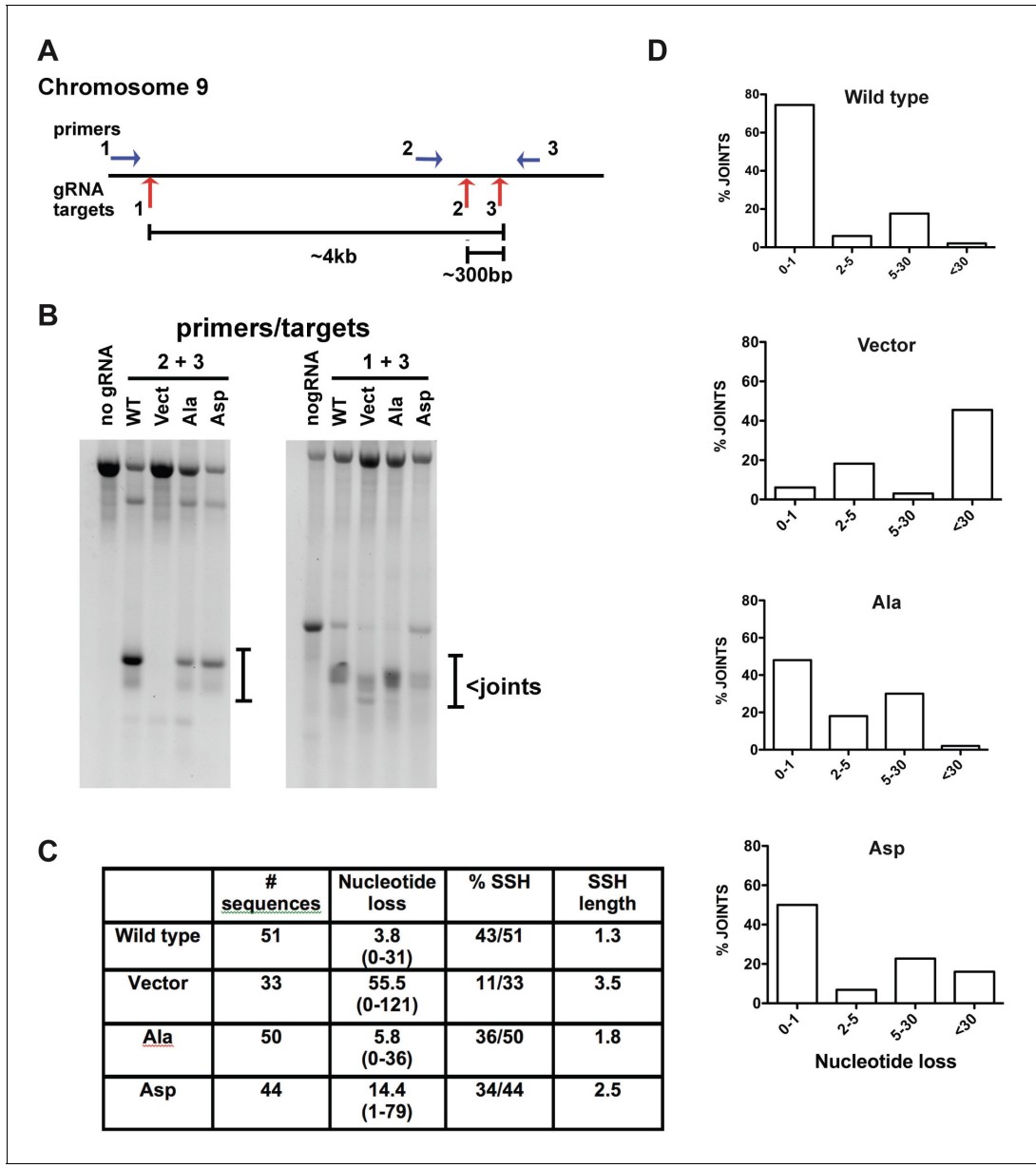

**Figure 13.** Phospho-mimicking XRCC4 and XLF alters repair of chromosomal DNA DSBs. (A) Diagram of region on chromosome nine targeted by three different gRNAs and position of primers utilized to detect chromosomal deletions induced by Cas9 and indicated gRNAs. (B) Ethidium bromide staining of PCR amplifications of DNA isolated from cells expressing WT, Ala, Asp, or no XRCC4 and XLF and transfected with the different combinations of gRNAs as indicated. A no gRNA control DNA of parental 293 T cells is also included to help define products induced by Cas9 induced deletions. (C) Summary of sequenced joints utilizing gRNAs 1 and 3 that result in a 4 kb chromosomal deletion. (D) Graphs represent the percentage of joints deleting increasing numbers of nucleotides from predicted double strand break site.

combinations of gRNA/Cas9 plasmids. Cells were harvested and DNA prepared 72 hr later and PCR performed to detect chromosomal deletions. As can be seen, chromosomal deletions are readily detected with both gRNA pairs in cells that have been reconstituted with XRCC4 and XLF. Although no joints are observed in the vector control cell strain with one gRNA pair, deletional joints are observed with the other suggesting that certain DSBs may be more readily joined by a-NHEJ than other DSBs. By electrophoretic mobility, the deletional joints using gRNAs 2 and 3 (that induce the small 300 bp deletion) appear remarkably similar in cells expressing WT, Ala, and Asp versions of XRCC4 and XLF (*Figure 13B*). Indeed, sequence analysis revealed no significant differences (data not shown). In contrast, the 4 kb deletions generated in the four cell strains appear much more heterogeneous. By sequence analyses, joints recovered from cells expressing WT or alanine substituted XRCC4 and XLF are very similar. As expected, joints recovered from cells lacking XRCC4 and XLF display increased nucleotide loss. Although joints from all four cell strains predominately occur at regions of short end terminal homology (SSH), the length of SSH is increased in cells lacking XRCC4 and XLF (*Figure 13C*). Joints recovered from cells expressing aspartate substituted XRCC4 and XLF show an intermediate increase in nucleotide loss, and moderate increase in the length of SSH utilized in joining (*Figure 13D*). We conclude that DNA bridging facilitates accurate repair of certain chromosomal DNA DSBs.

## Discussion

Here we report that blocking or mimicking DNA-PK/ATM phosphorylated residues in the disordered C-terminal tails of XRCC4 and XLF impacts on the stability of XRCC4/XLF complexes, which are readily dismantled from DNA when DNA-PK/ATM sites are phospho-mimicked. We propose that ATM/DNA-PK phosphorylation of XRCC4/XLF filaments facilitates their dissociation from DNA.

## Role of the disordered XRCC4 and XLF C-terminal tails in modulating the dynamics of XRCC4-XLF complexes

How could the disordered C-terminal tails of XRCC4 and XLF affect the behavior of XRCC4-XLF complexes at the molecular level? Structural and docking studies have revealed that XRCC4 and XLF can form alternating filamentous structures that can further assemble into bundles with a central channel (*Andres et al., 2012*; *Hammel et al., 2011*; *Ropars et al., 2011*; *Wu et al., 2011*). One model proposes that when interacting with DNA, XRCC4-XLF complexes would wrap around DNA, trapping it in the central channel forming a tube-like structure that can diffuse along the DNA with very little friction (*Brouwer et al., 2016*). DNA tethering by these bundles would be promoted by the ability of XRCC4 dimers to tetramerize by interdigitation of their α-helical stalks (*Andres et al., 2012*). However, this model doesn't take into account the positions of the disordered C-terminal tails of XRCC4 and XLF because the crystal structures were obtained with truncated forms of XRCC4 and XLF lacking the C-terminal regions. Yet, building upon this model and others (*Hammel et al., 2011*; *Ropars et al., 2011*; *Wu et al., 2011*), there are several potential mechanisms by which the disordered C-terminal tails of XRCC4 and XLF could stabilize the proteins/DNA interaction. First, the tails could promote stabilization of the XRCC4-XLF homodimer-homodimer interactions during filament assembly and concurrently favor DNA binding since the DNA binding domain of XLF resides at the extreme C-terminus (*Figure 1B*, *Andres et al., 2007*). Second, the tails could be involved in stabilization of filament bundles (*Andres et al., 2012*; *Wu et al., 2011*) that could correspond to the sleeve-like structure sliding on DNA that were observed at the single molecule level (*Brouwer et al., 2016*). Finally, the tails could also contribute to the robust DNA tethering observed for XRCC4-XLF complexes (*Brouwer et al., 2016*). Of note, in ITC experiments the thermodynamic parameters that contribute to the free energy of the XRCC4-XLF interaction (enthalpy and entropy) are distributed differently for the XRCC4-Asp-XLF-Asp complex as compared to either the WT-WT and Ala-Ala combinations. These data suggest that the tails may adopt a variety of conformations; deciphering these will require additional experimental strategies [nuclear magnetic resonance spectroscopy, electron paramagnetic resonance, FRET, and molecular dynamics simulations].

## Intrinsic XLF DNA binding and bridging activities

In vitro, XRCC4 and XLF independently bind to DNA with greater stability as the DNA substrate increases in size (*Hentges et al., 2006*; *Lu et al., 2007*; *Modesti et al., 1999*). Clearly, XRCC4-XLF

complexes bind to and bridge DNA with much higher affinity as compared to that of either homo-dimer alone (*Andres et al., 2012*). Here, we find that XLF alone is able to bridge DNA when the size of the DNA molecule is increased (2.7 kb instead of 0.5 kb, *Figure 7*). Given that XLF is a dimer where each monomer has a single DNA binding domain located within the very last ~20 residues of the C-terminus (*Andres et al., 2007*), we propose that a single XLF dimer interacts with, thus bridges, two independent DNA molecules. The XLF DNA binding domain is akin to the 'AT-hook' DNA binding motifs present in High Mobility Group A proteins; these AT-hook motifs are all similar to XLF's C-terminal tail in that they are intrinsically disordered DNA binding protein domains (*Aravind and Landsman, 1998*; *Fonfría-Subirós et al., 2012*). The AT-hook motif is a weak DNA binding module interacting with the minor groove of DNA preferentially with AT rich regions. If XLF shares this mode of DNA binding, this would provide a good explanation for the observed concentration and DNA size dependence that is requisite for stable DNA binding and bridging by XLF. In addition, we find the XLF-Asp phospho-mimetic variant, in which all the 'phosphorylated' residues are located in the disordered tails but away from the very C-terminal DNA binding domain, exhibits normal DNA binding by EMSA but has a reduced ability to bridge DNA (*Figures 3* and *7*). A possible explanation for this behavior is that one of the two XLF 'phosphorylated' tails in a homodimer masks the DNA binding domain of the other tail in the XLF dimer with sufficient affinity to confer, DNA binding, but not to facilitate bridging. Alternatively, they could attach and detach fast from DNA so that, overall, they are bound to DNA but not stably enough to bridge two DNA molecules.

## Inferences on the regulation of XRCC4 and XLF by phosphorylation

Several levels of phosphorylation-dependent regulation of XRCC4 and XLF have been reported that affect function. Phosphorylation of XRCC4 on T233 by CDK2 has been implicated in recruitment of PNK, Aprataxin, and APLF through interaction with the FHA domain present in these DNA processing enzymes, thus affecting the scaffold function of XRCC4 (*Cherry et al., 2015*; *Clements et al., 2004*; *Iles et al., 2007*; *Koch et al., 2004*; *Macrae et al., 2008*). Phosphorylation of XLF by Akt on T181 triggers its dissociation from the LIG4/XRCC4 complex and its sequestration in the cytoplasm (*Liu et al., 2015*). However, a functional impact for either of these phosphorylations on the actual efficacy of c-NHEJ has not been reported.

The functional relevance of DNA-PK/ATM phosphorylations of either XRCC4 or XLF is a long-standing question. Although DNA-PK phosphorylation of XRCC4 in vitro reduces its affinity for DNA (*Modesti et al., 1999*), neither XRCC4 nor XLF DNA-PK/ATM dependent phosphorylations are required for c-NHEJ as measured by complementation of radiosensitivity or V(D)J joining assays in living cells (*Lee et al., 2004*; *Yu et al., 2003*, *2008*). We have reported previously that DNA-PK auto-phosphorylation sites exist as a functional cluster; in these studies, ablation of one or two sites had no functional impact. The functional impact of phosphorylation could only be appreciated when most or all of the sites in the cluster were abrogated (*Cui et al., 2005*; *Ding et al., 2003*; *Neal et al., 2014*). Thus, if the C-terminal tails of XLF and XRCC4 function together, the impact of phosphorylation might not be apparent if each is studied in isolation, as in the approach utilized in previous studies to assess the functional relevance of XLF and XRCC4 DNA-PK/ATM mediated phosphorylations (*Yu et al., 2003*, *2008*). Thanks to the advent of CRISPR/Cas9 genome-editing techniques, the potential functional redundancy or the need for concomitant phosphorylation in multiprotein complexes can be more easily assessed. In fact, cells deficient in multiple factors can readily generated, paving the way to study the impact of phosphorylation of the C-termini of both XRCC4 and XLF in concert. Here, we show that simultaneously phospho-mimicking sites in both the C-terminal tails of XRCC4 and XLF, completely abrogates DNA tethering in vitro, suggesting no functional redundancy between these phosphorylations, and that phosphorylation of both is required to efficiently disrupt XRCC4-XLF complexes bound to DNA. Phospho-mimicking mutants are severely compromised in DNA bridging in vitro. Consistent with a functional role in DNA bridging of chromosomal DSBs, co-expression of XRCC4 and XLF phospho-mimetic variants in XRCC4/XLF double deficient 293 T cells, HCT116 cells, or XR-1 cells failed to fully rescue cellular sensitivity to neocarzinostatin — even though these complexes can stimulate LIG4/XRCC4 activity in vitro and join both RAG or I-Sce1-induced DSBs on episomal substrates in living cells. We conclude that DNA bridging is required to repair neocarzinostatin induced chromosomal DNA DSBs. This conclusion is bolstered by results presented in *Figure 13* demonstrating that rejoined chromosomal DSBs from cells that cannot form XRCC4/XLF filaments (because of phospho-mimicking substitutions) display

increased nucleotide loss, and a higher dependence on short sequence homology at the DNA termini.

Moreover, co-expression of XRCC4 and XLF phospho-blocking mutants in XRCC4/XLF double deficient cells only partially reversed sensitivity to neocarzinostatin providing evidence that phosphorylation-dependent disruption of XRCC4 and XLF complexes is also very important for repair. We suggest that dismantle of XRCC4-XLF-DNA complexes (after repair is complete) is a crucial step in c-NHEJ, perhaps analogously to the recently established requisite release of the Ku ring after completion of repair (*Brown et al., 2015*; *Postow, 2011*; *Postow et al., 2008*; *van den Boom et al., 2016*), and that is tightly regulated by phosphorylation of the disordered C-terminal tails of XRCC4 and XLF, possibly by DNA-PK/ATM.

## Materials and methods

### Recombinant protein expression constructs and purification

Expression plasmids and purification procedures for XRCC4-WT, XRCC4(1–157), XLF-WT, XLF(1-224), XLF-L115A, XLF-L115D have been described (*Andres et al., 2012*, *2007*). The expression constructs for the XRCC4 and XLF Ala and Asp variants were generated by site-directed mutagenesis of the vectors expressing the WT proteins as indicated in *Figure 1*, and purified by the methods indicated above. The LIG4/XRCC4 complex was produced as described (*Cottarel et al., 2013*). The LIG4/XRCC4-eGFP variants were produced by co-transformation of pCDFduet-1-LIG4-His6 and pRSFduet-1-XRCC4-eGFP (WT, Ala or Asp variants) in Rosetta 2 pLysS cells (Novagen), expressed and purified as indicated above for the LIG4/XRCC4 complex. All proteins batches were dialyzed against 150 mM KCl, 20 HEPES pH8, 1 mM EDTA, 2 mM DTT and 10% (v/v) glycerol, flash frozen in liquid nitrogen and stored at −80°C. T4 DNA ligase was obtained from New England Biolabs.

### EMSA

Binding reactions (10 µl) contained 200 ng of linearized pUC19 plasmid DNA, 10 mM HEPES pH 8, 0.5 mM EDTA, 0.5 mM DTT, 5% (v/v) glycerol, and 75 mM KCl after addition of proteins at the indicated concentrations. Reactions were incubated at room temperature for 30 min and directly resolved by 0.6% agarose gel eletrophoresis in standard Tris-Borate-EDTA buffer. Gels were stained in Tris-Borate-EDTA buffer supplemented with 0.5 µg/mL ethidium bromide, destained in deionized water and documented using a UV light trans-illuminator and a CCD camera.

### Ligation

DNA ligation assays were performed in a volume of 10 µL containing 200 ng of linearized pUC19 plasmid DNA (*Xba*I for cohesive ends and *Sma*I for blunt ended ends), 1 mM ATP, 2 mM MgCl₂, 10 mM HEPES pH 8, 0.5 mM EDTA, 1 mM DTT, 5% (v/v) glycerol, and 75 mM KCl after addition of proteins at the indicated concentrations. Reactions were incubated at room temperature for 15 min at room temperature before addition of 40 cohesive end units of T4 DNA ligase, or LIG4/XRCC4 (and XLF) at the indicated concentrations and further incubated for 30 min at room temperature. At the end of the incubation Sarkosyl (Sigma-Aldrich) and Pronase (Roche) were added at, respectively, 2% (w/v) and 1 mg/mL final concentrations and incubated for 30 min at 55°C. Samples were next supplemented with 1/5 vol of a solution containing 50 mM EDTA, 50 mM Tris pH 7.5, 60% (v/v) glycerol and tracking dyes, and resolved by 0.8% agarose gel electrophoresis in standard Tris-Borate-EDTA buffer. Gels were stained in Tris-Borate-EDTA buffer supplemented with 0.5 µg/mL ethidium bromide, destained in deionized water, and documented using a UV light trans-illuminator and a CCD camera.

### Bridging assay

The one-end biotinylated 1000 bp DNA substrate was prepared by PCR using Phire Hot Start DNA Polymerase (New England Biolabs), primers 5'-Biotin-GAGTTTTATCGCTTCCATGACG and 5'-AA TTTATCCTCAAGTAAGGGGC and φX174 DNA as template. The blunt ended 500 bp DNA fragment was similarly prepared but using primers 5'-GAGTTTTATCGCTTCCATGACG and 5'-CAGAAAA TCGAAATCATCTTCG. The blunt ended 2.7 kb DNA substrate was prepared by digesting of pUC19 DNA with *Sma*I, gel purified, and stored in 10 mM Tris pH 8, 1 mM EDTA. For the DNA bridging

assay, 20 µL of streptavidin-coated bead suspension (Dynabeads M-280 Streptavidin, Invitrogen) were washed twice with 100 µL of DNA binding buffer (DBB) containing 150 mM KCl, 20 mM HEPES pH 8, 1 mM EDTA, 1 mM DTT, 10% (v/v) glycerol, 0.5 mg/mL of Acetylated BSA (Ambion) and 0.2% (v/v) TWEEN 20. Beads were resuspended in 20 µl DBB containing 1 µg of 1000 bp biotin-labeled DNA and incubated 10 min at 4°C to allow attachment. After washing twice with 100 µL DBB, beads were resuspended in 40 µL of DBB containing 1 µg of 500 bp (or 2.7 kb) DNA and the proteins at the indicated concentrations and incubated for 15 min at RT. Beads were next washed twice with 100 µL DBB and resuspended in 20 µL DBB without BSA but containing pronase at 1 mg/mL and 2% (w/v) Sarkosyl, and incubated at 55°C for 30 min. The samples were fractionated by 0.8% agarose gel electrophoresis in standard Tris-Borate-EDTA buffer. Gels were stained in Tris-Borate-EDTA buffer supplemented with 0.5 µg/mL ethidium bromide, destained in deionized water, and documented using a UV light trans-illuminator and a CCD camera.

## Protein crosslinking

XRCC4 and XLF were mixed each at 20 µM in 20 µL of 150 mM KCl, 20 HEPES pH 8, 1 mM EDTA, 2 mM DTT and 10% (v/v) glycerol incubated for 20 min at 4°C before addition of 20 µL of a stock of BS$^3$ at 12.5 mM in water (ThermoFisher Scientific). Reactions were incubated at room temperature for 1 hr and terminated by addition of 8 µL of 10% (w/v) SDS, 10 mM 2-mercaptoethanol, 20% (v/v) glycerol, 0.2 M Tris pH 6.8% and 0.05% (w/v) bromophenol blue. Samples were boiled for 3 min, fractionated by denaturing SDS polyacrylamide gel electrophoresis and the gels stained with Coomassie Brilliant Blue R250 before image capture.

## Isothermal titration micro-Calorimetry

Purified XRCC4 and XLF (WT, Ala and Asp variants) were dialysed overnight and diluted in a buffer consisting of 20 mM Tris pH 8 and 150 mM NaCl. Titration experiments were carried out on a Micro-Cal ITC200 micro-calorimeter (GE Healthcare, Piscataway, NJ). Experiments consisted of 13 injections of 3 µl of the titrant (XLF) into the analyte (XRCC4), using a titrant concentration set to be in the order of 10 or 13 times the analyte concentration. A first small injection (0.4 µl) was included in the titration protocol in order to remove air bubbles trapped in the syringe prior to the titration. Raw data were adjusted setting the zero to the titration saturation heat value. Integrated raw ITC data were fitted using a one site non-linear least squares fit model using the MicroCal plugin as available in Origin 9.1 (OriginLab Corp., RRID:SCR_002815). $K_d$, $\Delta G^0$, and $-T\Delta S^0$ values were calculated from the fitted $\Delta H^0$ and $K_a$ values using the relations:

$$K_d = 1/K_a$$

$$\Delta G^0 = -R \cdot T \cdot \ln(K_a)$$

$$\Delta G^0 = \Delta H^0 - T\Delta S^0$$

where $R$ is the ideal gas constant ($1.987 \, \text{cal} \cdot \text{K}^{-1} \cdot \text{mol}^{-1}$) and $T$ the absolute temperature

## Surface plasmon resonance

Surface Plasmon Resonance experiments have been conducted on a Biacore T200 instrument (GE Healthcare) using a four channels streptavidin-coated sensor chip (BR-1005–31, GE Healthcare). The chip was activated before DNA capture by a rapid wash with 1 M NaCl and 50 mM NaOH in MilliQ water and then in 50% (v/v) Isopropanol. The 400 bp-long dsDNA molecules have been generated by PCR using φX174 DNA as template and primers 5'-Biotin-GAGTTTTATCGCTTCCATGACG and 5'-ACTTGACTCATGATTTCTTACC, gel purified, diluted to a concentration of 22.5 ng/mL in 10 mM Tris pH 8, 100 mM KCl, and 1 mM EDTA, and flushed in channels 2, 3, and four until the instrument response reached respectively 258 RU, 252 RU, and 511 RU. Channel one was not flushed with the DNA and used as reference channel during data analysis. All proteins (WT and variants) have been diluted in 75 mM KCl, 20 mM HEPES pH 8, 10% (v/v) glycerol, 1 mM DTT, 1 mM EDTA to the desired final concentrations. The same buffer with the addition of 0.05% (v/v) TWEEN 20 was used as running buffer during experiments. Protein concentration was adjusted at about 10 µM, re-

inspected by NanoDrop to double check the initial concentration, and finally diluted to the desired final concentration. In kinetics experiments requiring protein titrations, subsequent dilutions were done using the same pipetting volume and a single pipette tip for each different protein dilution serie. Protein samples were injected in the chip channels and let interact with the DNA on the surface typically for 240 or 300 s. Dissociation was observed during 480 s before the channels were washed and the chip regenerated with a 30 s-long flush of a 2 M KCl aqueous solution. A SDS-PAGE gel of the protein samples used for SPR experiments is shown in *Figure 1—figure supplement 1*. In all cases with the exception of XRCC4 alone, a simple kinetics model of first order reactions failed to properly describe the data. We thus used equilibrium affinity and biphasic dissociation rates to quantify and compare the results.

## Cell lines

293 T cells (RRID:CVCL_0063) were the generous gift of Dr. Kefei Yu who purchased the cell strain from ATCC. XR-1 cells (RRID:CVCL_K253) were the generous gift of Dr. Tom Stamato. HCT116 cells (RRID:CVCL_0291) and the single XRCC4 (RRID:CVCL_HE00) and XLF (RRID: CVCL_HD86) knockout cells were the generous gift of Dr. Eric Hendrickson. Cells were regularly checked for mycoplasma contamination using the Mycoplasma Plus PCR primer set (Stratagene Cat# 302008).

## Antibodies

Antibodies used are mouse monoclonal anti-$\beta$-actin (sigma A5441, RRID:AB_476744); mouse monoclonal anti-$\gamma$-H2AX (Millipore 05–636 RRID:AB_309864); rabbit polyclonal anti-XLF (Abcam ab33499, RRID:AB_778945); rabbit polyclonal anti-XRCC4 (Serotec AHP387, RRID:AB_2218607).

## Cas9-mediated gene disruption

Cas9-targeted gene disruption was performed using methods similar to those reported by (*Mali et al., 2013*). Briefly, gRNAs specific for XLF, XRCC4 or ATM were synthesized as 455 bp fragments (Integrated DNA Technologies). The synthesized fragments were cloned into pCR2.1 using a TOPO TA cloning kit according to the manufacturers' instructions (Life Technologies). Cells were transfected with 1 µg gRNA plasmid and 1 µg Cas9 expression plasmid (Addgene, RRID:SCR_002037). In some cases, cells were co-transfected with 0.2 µg of pcDNA6 (Life Technologies) or pSuper-Puro to confer blasticidin or puromycin resistance. With the two human cell strains, western blotting was used to identify clones with deletions in each of these factors; in all cases, deletion was also confirmed by PCR amplification that revealed deletions at the target site. PCR amplification of the hamster XLF gene was used to define frameshift mutations in the XR-1 cell strain. The 19-mers specific for each factor synthesized into the 455 bp fragments are as follows:

    ATM: TCTTTCTGTGAGAAAATAC
    XRCC4: CCTGCAGAAAGAAAATGAA
    XLF: GGCCTGTTGATGCAGCCAT.

## Plasmid constructs for cellular assays

The expression constructs providing for co-expression of WT, Ala, and Asp XRCC4 and XLF were sequentially assembled into pMSCV-puro (Clontech), adding C-terminal Myc tags to each protein, and an IRES sequence between XRCC4 and XLF. Phosphorylation site substitutions are indicated in *Figure 1*. RAG1 and RAG2 expression plasmids were a gift from Dr. David Roth. Construction of the I-SceI expression plasmid was described (*Neal et al., 2016*). The I-Sce1-trex2 fusion expression plasmid was obtained from Dr. Jeremy Stark. Fluorescent V(D)J substrate plasmids have been described (*Neal et al., 2016*). Briefly, the RFP coding sequence was inserted between the 12RSS and 23RSS, replacing the oop transcription terminator in both pJH290 and pJH289. The recombination cassette was cloned upstream of the CFP coding sequence in peCFP-N1 (which includes the SV40 origin of replication).

## Episomal end joining assays

The fluorescent V(D)J substrates (diagrammed in *Figure 10*, top panels) and V(D)J assays have recently been described (*Neal et al., 2016*). These substrates (derived from pECFP-N1, Clontech) contain the SV40 origin of replication, and are thus efficiently replicated episomally in all primate cell strains.

Briefly, extrachromosomal fluorescent V(D)J assays were performed on cells plated at 20–40% confluency into 24-well plates in complete medium. Cells were transfected with 0.125 µg substrate, 0.25 µg RAG1 and 0.25 µg RAG2 per well using polyethylenimine (PEI, Polysciences) using 2 µL of PEI at 1 µg/mL per 1 µg of DNA. In experiments with additional expression plasmids, 0.25 µg of the expression plasmid or vector control was included. Cells were harvested 72 hr after transfection and analyzed for CFP and RFP expression by flow cytometry. The percentage of recombination was calculated as the percentage of cells expressing CFP divided by the percentage expressing RFP. Data presented represent at least three independent experiments, which each includes triplicate transfections.

## Chromosomal end joining assays

For chromosomal joining assays, the plasmid pSpCas9(BB)—2A-Puro (Addgene, RRID:SCR_002037) which provides co-expression of a gRNA, Cas9, and the puromycin resistance gene was utilized. The target region 20-mers specific for three sites on chromosome nine are as follows:

    Target 1: GCTCCAGCTGCCTGGACCTG
    Target 2: GGGCCAGGCCTGGGTTCAAC
    Target 3: GACTTAAGAGAAAGGGACTG.

293 T cells expressing WT, Ala, Asp, or no XRCC4 or XLF were transfected with combinations of gRNAs as indicated and plated on puromycin after 24 hr. Cells were harvested 72 hr after transfection and genomic DNA extracted using DNAzol (Thermo-Fisher) according to the manufacturer's recommendations. PCR amplification was performed using Go-Taq (Thermo-Fisher) with the following amplification primers:

    Primer 1: CAGAAGTGCCCAGGCCCCTCC
    Primer 2: CCCAAGATGTCCCGGCTGTGGG
    Primer 3: CCATGGGCCTCTCTGTCCTTGCAC.

## Clonogenic survival assays

Clonogenic survival assays were performed for HCT116 cells and XR-1 cells. Briefly, a hundred cells were plated for each transfection into complete medium containing the indicated dose of zeocin or neocarzinostatin in 60 mm diameter tissue culture dishes. After 7 to 10 days, cell colonies were stained with 1% (w/v) crystal violet in ethanol to measure relative survival.

MTT staining was performed to assess cell viability for 293 T cells. 30,000 to 50,000 cells were plated in each well of a 24-well plate, containing medium with varying concentrations of drug. After 5 to 7 days of zeocin or neocarzinostatin treatment, cells were treated with 1 mg/mL MTT (Sigma) solution for 1 hr. Medium containing MTT was then removed and formazan crystals thus produced were solubilized in acidic isopropanol. Absorbance was read at 570 nm to determine relative survival.

## Acknowledgements

We thank Satish K Tadi and Bertrand Llorente for critical reading of the manuscript. This work was supported by Public Health Service grant AI048758 (KM), by grant PLBIO13-099 from the French National Cancer Institute (MM), Project n° SFI20121205867 from the ARC Foundation (MM).

## Additional information

### Funding

| Funder | Grant reference number | Author |
| --- | --- | --- |
| Fondation ARC pour la Recherche sur le Cancer | SFI20121205867 | Mauro Modesti |
| U.S. Public Health Service | AI048758 | Katheryn Meek |
| Institut National Du Cancer | PLBIO13-099 | Mauro Modesti |

The funders had no role in study design, data collection and interpretation, or the decision to submit the work for publication.

## Author contributions
DN, Investigation, Writing—review and editing; AN, AJdM, Investigation; SB, Methodology; KM, MM, Conceptualization, Supervision, Funding acquisition, Investigation, Writing—original draft, Writing—review and editing

## Author ORCIDs
Davide Normanno, http://orcid.org/0000-0003-4740-5542
Mauro Modesti, http://orcid.org/0000-0002-4964-331X

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
