## [Decision Letter]

Thank you for submitting your article "Phosphorylation modulates XRCC4-XLF-mediated DNA bridging" for consideration by *eLife*. Your article has been favorably evaluated by Jessica Tyler (Senior Editor) and three reviewers, one of whom is a member of our Board of Reviewing Editors. The reviewers have opted to remain anonymous.

The reviewers have discussed the reviews with one another and the Reviewing Editor has drafted this decision to help you prepare a revised submission.

Summary:

XLF and XRCC4 are thought to function, at least in part, by forming a filament that bridges two DNA ends prior to joining by NHEJ. The dissociation of this filament from DNA after joining could occur passively or actively. In this manuscript, the authors explore the notion that phosphorylation of XRCC4 and XLF is required to promote filament disassociation from DNA. To this end, they use a variety of biochemical assays that rely entirely on the comparison of combinations of XRCC4 and XLF proteins where potential ATM/DNAPK- phosphorylation sites have been mutated to alanine (phospho-blocking) or aspartate (phospho-mimicking). Most of the studies are carried out in vitro and are generally well done and convincing with respect to defects in the phospho-mimicking mutants ability to bridge and stabilize broken DNA ends. However, the veracity of these findings and their relevance to NHEJ in vivo are challenged a bit by some important inconsistencies.

Essential revisions:

1) The physiological relevance of all of the different potential ATM/DNA-PKcs phosphorylation sites analyzed in XRCC4 and XLF have not been fully evaluated. Whether these sites are phosphorylated in vivo is not known. The full replacement to Ala or Asp, could affect the overall structure or binding of the C-terminal tails independent of the potential effect of these mutations on phosphorylation. This important issue needs to be clearly presented and better discussed in the manuscript, including the title, which mentions phosphorylation even though phosphorylation per se is never examined.

2) The authors used BS3 cross-linking assay (Figure 4) to demonstrate the XRCC4 and XLF phospho-blocking and phospho-mimicking mutants do not interfere with protein-protein interactions. Cross-linking assays can capture weak, unstable and/or transient protein-protein interaction. It would be important to evaluate the strength and stability of the interaction between wild type and mutant XRCC4 and XLF proteins. If the interaction between the phospho-mimicking XRCC4 and XLF is weak this would explain why these proteins do not bridge DNA ends well. The authors should perform a traditional co-IP or pull down assay to compare the interaction between XLF and XRCC4 when they are both wild type and when they have phospho-blocking or phospho-mimicking mutations.

3) Using SPR assay (Figure 7), the authors conclude that XRCC4-XLF phospho-mimicking mutants dissociate much faster than the wild type or phospho-blocking mutants, implying that phosphorylation of the C-termini of XLF and XRCC4 regulates the disassembly of the XRCC4-XLF filaments. However, the data presented in Figure 7 shows that the phospho-mimicking mutants also binds DNA much less well, consistent with the notion that a primary effect of these mutations may be on assembly. This data is inconsistent with the data in Figure 3.

4) The phospho-mimicking XRCC4 and XLF do not enhance T4 DNA ligase activity as well as wild type XRCC4 and XLF, but they do promote DNA Ligase 4 activity equally well as the wild type protein. Based on the conclusions that the authors make about the function of the phosphorylation sites it is unclear why there should be this difference.

5) The lack of impact of the XRCC4-XLF phospho-mimicking mutants on VDJ recombination (Figure 9) is surprising since the final read-out requires transcription of the rejoined product. It might be expected that failure to dismantle the repair complex would impede that. In addition, the potential effect of the XRCC4-XLF phospho-mimicking mutants on the response to DNA damaging agents (Figure 10) is inconsistent between cell lines and DNA damaging agents making these findings inconclusive. Thus, the manuscript is lacking important in vivo data to support the notion that phospho-mimicking XRCC4 and XLF mutants have an effect on important events that occur after DSB joining during NHEJ.

Perhaps examining the resolution of g-H2AX foci would demonstrate changes in the resolution of the DNA damage response. XRCC4 and XLF are recruited to laser tracks and subsequently released. Such an approach may provide evidence for prolonged (phosphor-ablating mutants) or diminished (phosphor-mimicking mutants) presence at damage sites in live cells. The authors suggest that the requirement for dissociation of XRCC4 and XLF may be most important in the context of chromosomal DSBs, explaining why the episomal VDJ recombination assay did not show an effect. This could certainly be the case as there are examples of other requirements that are observed in the context of chromosomal DSB repair but not episomal DSB repair. There are cell-line based assays for chromosomal DSB repair that could be used to directly test this important notion.

By whatever means, some compelling evidence that the mechanism they are studying in vitro functions in vivo would be very helpful.

---

## [Author Response]

*Essential revisions:*

*1) The physiological relevance of all of the different potential ATM/DNA-PKcs phosphorylation sites analyzed in XRCC4 and XLF have not been fully evaluated. Whether these sites are phosphorylated in vivo is not known. The full replacement to Ala or Asp, could affect the overall structure or binding of the C-terminal tails independent of the potential effect of these mutations on phosphorylation. This important issue needs to be clearly presented and better discussed in the manuscript, including the title, which mentions phosphorylation even though phosphorylation per se is never examined.*

We have re-focused the manuscript on the role of the XRCC4 and XLF C-terminal tails via mutational phospho-mimicry analysis and have avoided direct claims about phosphorylation per se. We have clarified the presentation of our analysis and the manuscript in general. We have better and more carefully discussed our data. The title was changed accordingly. All the edits appear in red in the revised version of the manuscript.

*2) The authors used BS3 cross-linking assay (Figure 4) to demonstrate the XRCC4 and XLF phospho-blocking and phospho-mimicking mutants do not interfere with protein-protein interactions. Cross-linking assays can capture weak, unstable and/or transient protein-protein interaction. It would be important to evaluate the strength and stability of the interaction between wild type and mutant XRCC4 and XLF proteins. If the interaction between the phospho-mimicking XRCC4 and XLF is weak this would explain why these proteins do not bridge DNA ends well. The authors should perform a traditional co-IP or pull down assay to compare the interaction between XLF and XRCC4 when they are both wild type and when they have phospho-blocking or phospho-mimicking mutations.*

To address this point, we have analyzed the protein-protein interaction between XRCC4 and XLF by Isothermal Titration micro-Calorimetry, which provides a more quantitative analysis. The new results are reported in Figure 5, Table 1 and Figure 5—figure supplement 1. The results indicate that (XRCC4-WT + XLF-WT), (XRCC4-Ala + XLF-Ala) and (XRCC4-Asp + XLF-Asp) all interact with similar Kds. However, a difference was detected in the case of the (XRCC4-Asp + XLF-Asp) interaction where the contribution of the enthalpic and entropic thermodynamic parameters is distributed differently than in the case of the (XRCC4-WT + XLF-WT), and the (XRCC4-Ala + XLF-Ala) interactions. Such difference is often interpreted as related to protein rearrangements including conformation changes and accessibility of interaction surfaces upon displacement of solvent molecules.

*3) Using SPR assay (Figure 7), the authors conclude that XRCC4-XLF phospho-mimicking mutants dissociate much faster than the wild type or phospho-blocking mutants, implying that phosphorylation of the C-termini of XLF and XRCC4 regulates the disassembly of the XRCC4-XLF filaments. However, the data presented in Figure 7 shows that the phospho-mimicking mutants also binds DNA much less well, consistent with the notion that a primary effect of these mutations may be on assembly. This data is inconsistent with the data in Figure 3.*

The data in Figure 7 (now Figure 8) show that the equilibrium affinity is reduced 4-fold for the phospho-mimicking mutant combination (XRCC4-Asp + XLF-Asp) relative to the WT counterparts. The equilibrium affinity represents the ratio the equilibrium between the kinetic constants (C*k_on_/k_off_). However, we see find that the k_off_ for the phospho-mimicking mutant combination (XRCC4-Asp + XLF-Asp) is 4-times faster relative to the WT counterparts. Therefore, the drop in equilibrium affinity seenfor the phospho-mimicking mutant combination (XRCC4-Asp + XLF-Asp) relative to the WT counterparts can be accounted for by the increase in dissociation while the association remains unaltered. Unfortunately, the XRCC4-XLF interaction with DNA (filament polymerization) is not a simple 1:1 binding reaction and thus cannot be quantified by first order rate constants to directly evidence a change in the association and dissociation rates. This is now better explained in the text and is in line with the data shown in Figure 3, which are EMSA data that inform on apparent Kds and not kinetic parameters.

*4) The phospho-mimicking XRCC4 and XLF do not enhance T4 DNA ligase activity as well as wild type XRCC4 and XLF, but they do promote DNA Ligase 4 activity equally well as the wild type protein. Based on the conclusions that the authors make about the function of the phosphorylation sites it is unclear why there should be this difference.*

Stimulation of DNA ligation by XRCC4-XLF complexes is mechanistically different than stimulation of the LIG4/XRCC4 complex ligation activity by XLF.

XRCC4-XLF complexes stimulate DNA ligation through their ability to robustly tether DNA thereby favoring the rate of molecular collisions of the DNA ends during ligation (Andres et al., 2012, Brouwer et al., 2016). XRCC4-XLF complexes will therefore stimulate any ligase as shown in Figure 14 with the Chlorella virus DNA ligase.

Author response image 1.**DOI:**
http://dx.doi.org/10.7554/eLife.22900.034

The way in which XLF stimulates LIG4/XRCC4 is associated with an increased in the rate of re-adenylation of LIG4 but the mechanism of this stimulatory effect is not understood (Riballo E, Woodbine L, Stiff T, Walker SA, Goodarzi AA, Jeggo PA. XLF-Cernunnos promotes DNA ligase IV-XRCC4 re-adenylation following ligation. Nucleic Acids Res. 2009 Feb;37(2):482-492.). XLF could, by interacting with XRCC4 in the LIG4/XRCC4 complex, induce an allosteric change in the LIG4 catalytic site.

Our goal was to test whether the XRCC4-XLF phospho-mimicking mutants (that are affected in DNA bridging and therefore in their ability to stimulate a DNA ligation reaction) would affect human LIG4 activity. We therefore purified the LIG4/XRCC4 complex containing XRCC4-WT or XRCC4-Ala or XRCC4-Asp and tested the ability of XLF-WT or XLF-Ala or XLF-Asp to stimulate LIG4. In Figure 9 it can be seen that the XRCC4 and XLF phospho-mimicking or phosphor-ablation variants do not appreciably affect LIG4 activity. Thus, the XRCC4 and XLF phosphor-mimicking variants affect the ability of XRCC4-XLF complexes to bridge DNA but not LIG4 activity.

*5) The lack of impact of the XRCC4-XLF phospho-mimicking mutants on VDJ recombination (Figure 9) is surprising since the final read-out requires transcription of the rejoined product. It might be expected that failure to dismantle the repair complex would impede that. In addition, the potential effect of the XRCC4-XLF phospho-mimicking mutants on the response to DNA damaging agents (Figure 10) is inconsistent between cell lines and DNA damaging agents making these findings inconclusive. Thus, the manuscript is lacking important in vivo data to support the notion that phospho-mimicking XRCC4 and XLF mutants have an effect on important events that occur after DSB joining during NHEJ.*

*Perhaps examining the resolution of g-H2AX foci would demonstrate changes in the resolution of the DNA damage response. XRCC4 and XLF are recruited to laser tracks and subsequently released. Such an approach may provide evidence for prolonged (phosphor-ablating mutants) or diminished (phosphor-mimicking mutants) presence at damage sites in live cells. The authors suggest that the requirement for dissociation of XRCC4 and XLF may be most important in the context of chromosomal DSBs, explaining why the episomal VDJ recombination assay did not show an effect. This could certainly be the case as there are examples of other requirements that are observed in the context of chromosomal DSB repair but not episomal DSB repair. There are cell-line based assays for chromosomal DSB repair that could be used to directly test this important notion.*

*By whatever means, some compelling evidence that the mechanism they are studying in vitro functions in vivo would be very helpful.*

There is a long history of variable phenotypes in cells that lack XLF; in our previous work (Roy et al. 2015) we documented this variability in different cell strains. The HCT116 cell strain (although very popular) has several inherent issues (low expression of ATM, defect in one copy of MRE11, MMR defect); in fact, this cell strain is extremely sensitivity to zeocin [doses utilized in HCT116 transfectants are more than 10 folds lower than those used for 293T cells]. In contrast, the neocarzinostatin doses for both cell strains are the same. Thus, in three panels of transfectants in different cell types, the response to neocarzinostatin with wild type, Ala, Asp, and vector are all similar, and we conclude that XRCC4/XLF filaments (blocked in the Asp mutant) are required for full neocarzinostatin resistance, and that dissociation of XRCC4/XLF filaments (blocked in the Ala mutant) is also required for full neocarzinostatin resistance. To emphasize this point, in the current version of the manuscript, the HCT116 data, and XR-1 data are presented as supplemental data. The outlaying zeocin result is discussed, but not presented.

To document substantial blockade of XRCC4 phosphorylation by alanine substitution, an additional panel has been added to Figure 12 (previously Figure 10) showing ablation of the substantial electrophoretic mobility shift observed in wild type XRCC4 after damage induced phosphorylation. This supports the fact that the sites studied in this manuscript are authentic in vivo phosphorylation sites, as has been reported previously. In this same experiment, H2AX phosphorylation was also assessed and found to be similar in the isogenic cell strains.

We suggest that the lack of effect in episomal joining assays reflects a lack of requirement for XRCC4/XLF mediated DNA bridging. This is consistent with the in vitro results demonstrating that the Ala and Asp mutants are fully capable of stimulating LIG4 activity. In the current submission, we have also included a chromosomal end-joining assay (of gRNA mediated DSBs) in the isogenic 293T cell strains (Figure 13). As can be seen, repaired DSBs isolated from cells that cannot form XRCC4/XLF filaments (the Asp mutants) display increased nucleotide loss and a higher dependence on terminal microhomology at the DNA ends.